# Monocular Dynamic Gaussian Splatting: Fast, Brittle, and Scene Complexity Rules

**Yiqing Liang**[1], **Mikhail Okunev**[1], **Mikaela Angelina Uy**[2,3],
**Runfeng Li**[1], **Leonidas Guibas**[2], **James Tompkin**[1], **Adam W. Harley**[2]

[1]**Brown University**    [2]**Stanford University**    [3]**NVIDIA**

{yiqing_liang, mikhail_okunev, runfeng_li, james_tompkin}@brown.edu,
{mikacuy, guibas, aharley}@cs.stanford.edu

Reviewed on OpenReview: https://openreview.net/forum?id=fzmw8Joug4

## Abstract

Gaussian splatting methods are emerging as a popular approach for converting multi-view image data into scene representations that allow view synthesis. In particular, there is interest in enabling view synthesis for *dynamic* scenes using only *monocular* input data—an ill-posed and challenging problem. The fast pace of work in this area has produced multiple simultaneous papers that claim to work best, which cannot all be true. In this work, we organize, benchmark, and analyze many Gaussian-splatting-based methods, providing apples-to-apples comparisons that prior works have lacked. We use multiple existing datasets and a new instructive synthetic dataset designed to isolate factors that affect reconstruction quality. We systematically categorize Gaussian splatting methods into specific motion representation types and quantify how their differences impact performance. Empirically, we find that their rank order is well-defined in synthetic data, but the complexity of real-world data currently overwhelms the differences. Furthermore, the fast rendering speed of all Gaussian-based methods comes at the cost of brittleness in optimization. We summarize our experiments into a list of findings that can help to further progress in this lively problem setting.

## 1 Introduction

Let us consider the problem of monocular dynamic view synthesis. Using monocular camera data rather than multi-view data makes the problem highly ill-posed. In the past year, there have been hundreds of research papers on Gaussian splatting, with some applied to monocular dynamic view synthesis. Many papers claim superior performance compared to their peers even with only minor methodological differences between them—this seems implausible. The lack of a shared evaluation benchmark and inconsistent dataset splits make fair comparison between methods difficult, if not impossible, based solely on existing published results.

Our work provides a clear snapshot of progress on the problem of dynamic view synthesis for monocular sequences with Gaussian splatting approaches. Inspired in part by DyCheck (Gao et al., 2022), which benchmarked Neural Radiance Fields (NeRFs), our work quantifies the performance of existing work in apples-to-apples comparisons and provides analysis on failure modes. To support this analysis, we aggregate datasets used across current works, totaling 50 scenes. We also quantify performance on an instructive synthetic dataset that we created, which contains sequences with controlled camera and scene motion.

Our analysis establishes several findings that help clarify progress in this rapidly-moving field:

1. Gaussian methods struggle compared to hybrid neural fields: While Gaussian methods achieve fast rendering (20–200 FPS vs 0.3 FPS for TiNeuVox (Fang et al. (2022))), they are consistently outperformed by TiNeuVox in image quality across datasets and have similar (and more variable) optimization times.

2. Low-dimensional motion representations help: Local, low-dimensional representations of motion perform better than less-constrained systems, with field-based methods (DeformableGS (Yang et al. (2023)) & 4DGS (Wu et al. (2023))) showing better quality than per-Gaussian motion models.

3. Dataset variations overwhelm methodological differences: Contrary to claims in individual works, we find no clear rank-ordering of methods across datasets, with performance varying significantly by scene.

4. Adaptive density control causes significant issues: This mechanism leads to varying efficiency across scenes, increased overfitting risk, and occasional optimization failures.

5. Lack of multi-view cues particularly hurts dynamic Gaussians: On strictly-monocular data like the iPhone (Gao et al. (2022)) dataset, all Gaussian methods perform substantially worse than TiNeuVox.

6. Narrow baselines and fast objects cause systematic errors: Our synthetic dataset reveals degraded reconstruction quality as camera baselines decrease or object motion increases.

7. Specular objects remain challenging for all methods: Contrary to some claims, all current methods struggle with scenes containing reflective surfaces.

8. Metrics can be dominated by static background regions, requiring specific measurement on dynamic regions through static/dynamic masks to quantitatively reveal progress in methods.

**Public release.** Code and qualitative results including our instructive synthetic dataset can be found here. Data can be downloaded here, including segmentation masks and better camera poses for existing datasets.

## 2 Overview

First, we will establish the specific background relevant to our experiments. We present a more comprehensive discussion of related work in our supplementary material.

### 2.1 3D Gaussian Splatting (3DGS)

Gaussian splatting (Kerbl et al., 2023) represents a scene as a set of 3D anisotropic Gaussians allowing for high-quality and fast view synthesis given only input images for a static 3D scene. Concretely, the scene is represented as a collection of 3D Gaussians with underlying attributes, where each Gaussian $G_i = (x_i, \Sigma_i, \alpha_i, c_i)$ is parameterized with mean $x_i$, covariance matrix $\Sigma_i$, opacity $\alpha_i$, and view-dependent RGB color $c_i$ parameterized by 2nd-order spherical harmonic coefficients. The covariance matrix $\Sigma_i$ is factorized into a rotation matrix $R_i$ derived from a unit quaternion $q_i \in \mathbb{R}^4$, and scaling vector $s_i \in \mathbb{R}^3_+$ to guarantee it to be positive semi-definite, giving $\Sigma_i = R_i \text{diag}(s_i)\text{diag}(s_i)^T R_i^T$.

These parameters are optimized directly given a set of images with known camera intrinsics $K$ and extrinsics $V$ via gradient descent using differentiable rasterization. To form an image, the color of a pixel $(u, v)$ can be obtained by alpha-blending the contributions of the visible Gaussians $\{G_j\}_{j=1}^N$ sorted by depth along the camera ray for pixel $(u, v)$:

$$C(u,v) = \sum_{j \in N} c_j \alpha_j p_j(u,v) \prod_{k=1}^{j-1} (1 - \alpha_k p_k(u,v)) , \tag{1}$$

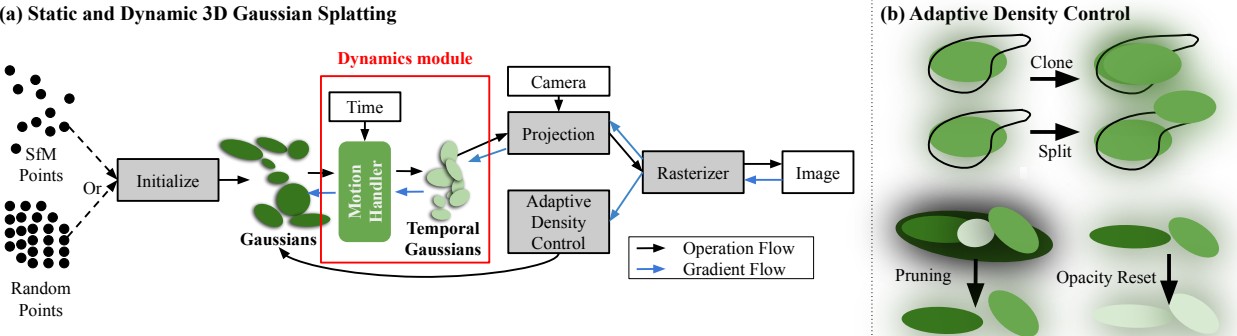

Figure 1: **Gaussian splatting with motion.** a) Overview of 3D Gaussian Splatting. Adding motion to static 3DGS typically modifies the pipeline only by adding a dynamics module, as marked by the red box. b) Adaptive densification via cloning, splitting, and pruning Gaussians to reflect scene details. This brittle process often causes difficulties during optimization for monocular dynamic sequences.

Table 1: **Overview of existing dynamic GS-based methods.** We organize and summarize the type, motion model and list of monocular datasets tested on by the different methods.

| Method | Ref. | Type | Motion model | Monocular; tested datasets |
|---|---|---|---|---|
| Dynamic 3D Gaussians | Luiten et al. (2023) | Iterative | - | ✗ - |
| 3DGStream | Sun et al. (2024) | Iterative | - | ✗ - |
| RTGS | Yang et al. (2024) | 4D | - | ✓ dnerf |
| Rotor-Based 4DGS | Duan et al. (2024) | 4D | - | ✓ dnerf |
| SpaceTimeGaussians | Li et al. (2023a) | 3D+motion | Poly+RBF | ✗ - |
| GauMesh | Xiao et al. (2024) | 3D+motion | HexPlane | ✗ - |
| GauFRe | Liang et al. (2023) | 3D+motion | MLP | ✓ dnerf, hypernerf, nerfds |
| Deformable-GS | Yang et al. (2023) | 3D+motion | MLP | ✓ dnerf, hypernerf, nerfds |
| MoDGS | Liu et al. (2024) | 3D+motion | MLP | ✓ - |
| 4DGS | Wu et al. (2023) | 3D+motion | HexPlane | ✓ dnerf, hypernerf |
| EffGS | Katsumata et al. (2023) | 3D+motion | Fourier+Poly | ✓ dnerf, hypernerf |
| DynMF | Kratimenos et al. (2023) | 3D+motion | Fourier+MLP | ✓ dnerf, hypernerf |
| Shape of Motion | Wang et al. (2024) | 3D+motion | Poly+MLP | ✓ iphone |
| Gaussian-Flow | Lin et al. (2023) | 3D+motion | Fourier+Poly | ✓ dnerf, hypernerf |
| GaGS | Lu et al. (2024) | 3D+motion | Voxel+MLP | ✓ dnerf, hypernerf |
| E-D3DGS | Bae et al. (2024) | 3D+motion | MLP | ✓ hypernerf |

where $p_j(u, v)$ is the contribution of Gaussian $G_j$ to pixel $(u, v)$, which is the probability density of the $j$-th Gaussian at pixel $(u, v)$. To obtain $p_j(u, v)$ given camera intrinsic $K$ and extrinsic $V$, the 3D Gaussian $G_j = \mathcal{N}(x_j, \Sigma_j)$ is approximated by a 2D Gaussian via linearization of the perspective transformation (Zwicker et al., 2001). This gives the 2D mean $x'_j = KV x_j \in \mathbb{R}^2$ and covariance $\Sigma'_j = JV\Sigma_j V^T J^T$, where $J$ is the Jacobian of the perspective projection.

To balance rendering quality and computational efficiency, 3DGS uses adaptive densification via periodically performing cloning, splitting, and pruning of the Gaussians (Figure 1).

The optimization requires the static scene to be observed from multiple viewpoints such that 3D Gaussians can converge to the underlying 3D scene geometry even under the ambiguity from 2D projection. This makes it extremely challenging for both the monocular and dynamic setting. Further difficulty is introduced in non-Lambertian scenes where view-dependent scene elements such as glossy objects or mirrors can cause Gaussians to be mistakenly placed to hallucinate reflections.

## 2.2 Gaussian Splatting for Dynamic View Synthesis

Extending 3DGS methods to dynamic scenes from monocular input requires deciding how to represent time $t$ (or spacetime). For instance, what kind of model is used to describe Gaussian motion, whether that model applies to Gaussians individually or collectively, whether motion is offset from a single timestep or from a canonical space representing all time, or even whether to use a motion model at all ('3D+motion') or simply to represent a 4D space (Tab. 1). Methods must also decide which Gaussian parameters to change over time, e.g., position, rotation, typically via offsets $\delta G_{i,t}$. The choice of motion design can significantly impact the expressiveness, efficiency, and robustness of the overall dynamic 3DGS system.

**Motion reference frame.** Iterative approaches assume that Gaussian motion over time can be optimized from some reference timestep (say, $t = 0$) in which Gaussians are already well-placed. Liuten et al. first showed multi-view dynamic 3DGS (Luiten et al., 2023) by updating Gaussian parameters one frame at a time. We might define a discrete function $f$ to query at each time $t$ to obtain Gaussian offsets:

$$f(i, 0) = G_i \ , \qquad f(i, t) = f[i, t-1] + \delta G_{i,t}. \tag{2}$$

This approach relies upon the reference frame being well reconstructed, such as from multi-view input, otherwise errors can be propagated across time (Fig. 2). Instead, canonical methods (Yang et al. (2023); Wu et al. (2023); Kratimenos et al. (2023); Liang et al. (2023); Bae et al. (2024); Lu et al. (2024)) assume 3D Gaussians are offset from an embedding that represents *all* of time without being any one frame. This helps for monocular cameras where multi-view constraints must be formed over time. Effectively, all monocular methods use some form of canonicalization.

**Motion complexity.** Motion may be defined by a function $f$ of time $t$ with varying complexity. For instance, we could define Gaussian motion to be linear over time, which would only be able to describe basic motions. A piecewise-linear model could explain many motions—e.g., Luiten et al.'s Dynamic 3D Gaussian work is effectively piecewise linear (Luiten et al., 2023) as Gaussian positions vary over integer timesteps. But, for monocular input, in practice authors must find low-dimensional 'sweet spot' functions that can constrain or smooth the ill-posed motion optimization.

Curve methods use a polynomial basis of order $L$ to define an $f$ over time that determines Gaussian offsets (Lin et al., 2023). $f$ could also use a Fourier basis (Katsumata et al., 2023) or a Gaussian Radial

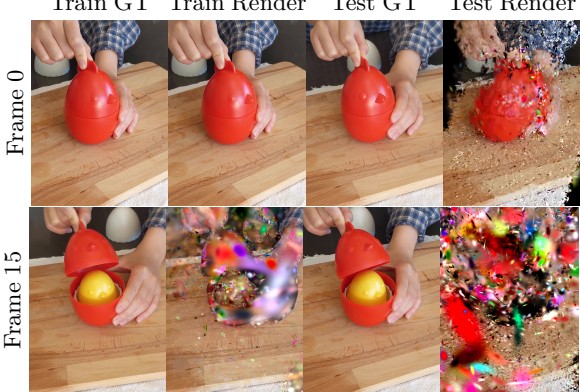

Train GT   Train Render   Test GT   Test Render

Frame 0

Frame 15

Figure 2: Iterative methods fail on monocular input.

Basis (Li et al., 2023a). Most of these typically use low order, where $L = 2$ or surprisingly even $L = 1$. We could also define a motion function with a multi-layer perceptron (MLP). A ReLU MLP would be a piecewise-linear model with changes at arbitrary points in continuous time rather than at integer times.

Motion complexity is important to reconstruction quality and computational speed. It also affects the success of optimization as it determines motion smoothness. For instance, MLP-based method are robust function fitters with self-regularization properties, but come with a higher computational burden. In our survey, we found that function complexity like the polynomial order was not always reported, making it difficult to consider these trade-offs.

**Motion locality.** As 3D Gaussian splatting is primitive based, many dynamic approaches define motion models also as a per-Gaussian property. This extends the parameter set of each Gaussian by the parameters of the motion model, which can be costly in terms of memory, and ignores the spatial implications of motion: that nearby primitives may move together.

Field-based approaches (Xie et al., 2022) attempt to encapsulate this idea by using a function to estimate the entire motion field. Gaussians then query this field by their position and time or by an embedding vector. Fields can be advantageous in that the number of parameters in the motion model is independent of the number of Gaussians, and that the field can enforce smoothness between Gaussians indirectly by the complexity of the motion function.

One popular approach is to use field embeddings to represent both locality in space and motion over time (Liang et al., 2023; Yang et al., 2023; Wu et al., 2023; Lu et al., 2024). For instance, a continuous deformation field encoded by an MLP $f_\theta$ might take as input each Gaussian $G_i$'s embedding $z_i$ and $t$ to produce an offset:

$$f_\theta(z_i, t) = \delta G_{i,t} , \qquad G_{i,t} = G_i + \delta G_{i,t} \tag{3}$$

As they represent volumes of continuous space, field-based approaches can exploit ideas from volume-based NeRF literature, such as voxel (Lu et al., 2024) and HexPlane (Wu et al., 2023) structures. These discrete data structures can also help to aggregate information and accelerate indexing.

Finally, intermediate approaches have begun to aggregate motion model parameters over local Gaussian neighborhoods (Lei et al., 2024). In general, there are many schemes to model both motion complexity and locality, but it helps to consider how both space and time are represented and smoothed.

**Higher dimensionality instead of motion.** Another alternative instead of modelling motion together with 3D Gaussians is to define Gaussians directly in 4D space (Yang et al., 2024; Duan et al., 2024): mean $x_i \in \mathbb{R}^4$ and covariance matrix $\Sigma_i \in \mathbb{R}^{4 \times 4}$. This representation couples the space ($\mathbb{R}^3$) and time ($\mathbb{R}$) dimensions. To optimize this representation through rendering, the 4D Gaussians are first conditioned on or projected to a given time $t$ to obtain 3D Gaussians; then, these can be rendered and compared to the 2D images at the given timeframes. 4D methods are flexible and can describe complex dynamic scenes. However, as there are many possible projections of plausible motions within the 4D space, these models may be difficult to fit correctly with the few constraints provided by monocular input. Moreover, the representation also does not have inherent smoothness or locality constraints.

# 3 Evaluation Setup

## 3.1 Video Datasets

Sequence variations affect performance significantly within and across datasets. Scenes with small scene motion only or large camera motion relative to object motion are easier than sequences with large scene motion and small camera motion. Scenes with highly-textured objects are easier to reconstruct but require more Gaussians. To gain as much variation as possible, we collect all common datasets used across papers. We briefly explain key differences here; our supplemental material provides a fuller description and examples.

D-NeRF (Pumarola et al. (2020)) shows synthetic objects captured by 360-orbit inward-facing cameras against a white background (8 scenes). Nerfies (Park et al. (2021a)) (4 scenes) and HyperNeRF (Park et al. (2021b)) (17 scenes) data contain general real-world scenes of kitchen table top actions, human faces, and outdoor animals. NeRF-DS (Yan et al. (2023)) contains many reflective surfaces in motion, such as silver jugs or glazed ceramic plates held by human hands in indoor tabletop scenes (7 scenes). Some existing datasets (D-NeRF, Nerfies, HyperNeRF) use a "teleporting" camera setting (Gao et al. (2022)) instead of using natural handheld camera trajectories: views from different cameras are used at different timesteps to create a pseudo-monocular video. This setup provides good multi-view constraints but is unrealistic: real handheld videos have more continuous camera motion and fewer viewpoint changes. NeRF-DS dataset's cameras do not "teleport", but the scene motion is small by design. The iPhone dataset from DyCheck (Gao et al. (2022)) (14 scenes) has dynamic objects undergoing large motions, with real-world camera trajectories.

In total, these comprise **50** scenes. For all datasets, we use the original train/test splits as provided by the authors to ensure fair comparison.

**Instructive synthetic dataset.** To vary scene motion complexity in a controlled way, we also create a dataset with varying camera and scene motion (Figure 3). Each scene is 60 frames long. These scenes contain a simple textured cube and a textured background wall. While simple, these scenes have both slow and fast camaera and scene motion, and many methods struggle to reconstruct them. In *SlidingCube*, the cube accelerates from stationary along a straight line to cover different distances $D \in \{0, 5, 10\}$. The camera moves around the cube in an arc with a total distance of $B \in \{1, 3, 5, 10, 20\}$. The camera rotates to track the cube at all times. In *RotatingCube*, the cube additionally rotates along its vertical axis by $\pi$ radians.

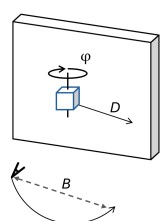 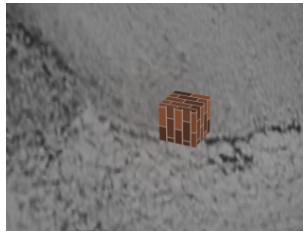 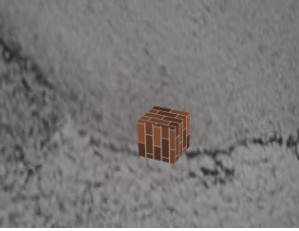

Figure 3: *Left:* Illustration of the instructive dataset. A simple textured cube moves in a straight line (distance $D$) against a differently-textured background. In another condition, the cube additionally performs an overall rotation ($\phi$ radians). The training-view camera moves in an arc (covering distance $B$) in the same direction as the cube, tracking it. The test camera (not depicted) travels in a sinusoidal trajectory behind the training camera. *Middle:* A training view. *Right:* A test view at the same time instance.

## 3.2 Metrics

For view synthesis, we use image quality metrics of peak signal-to-noise ratio (PSNR), Structural Similarity Index (SSIM) (Wang et al., 2004) and its variant Multi-Scale SSIM (MS-SSIM) (Wang et al., 2003), and learned perceptual metric LPIPS (Zhang et al., 2018) using the AlexNet-trained features. We also report training and rendering time in seconds. Existing research typically reports a metric as a mean average over sequences with one optimization attempt; we also report the standard deviation of three independent optimizations. All train time and rendering speeds are computed on NVIDIA GeForce RTX 3090 cards.

**Static vs. dynamic.** Static scene areas imaged by a moving camera may have enough constraints for reconstruction, and if these areas are large then they will dominate the metric. So, to demonstrate reconstruction quality on dynamic regions only, we compute per-frame binary dynamic masks using SAM-Track (Cheng et al., 2023) and report masked metrics mPSNR, mSSIM, mMS-SSIM and mLPIPS.

**Mean and Variance** For each experiment, we perform 3 runs to capture performance variability. In our visualizations, the height of each bar represents the mean ($\mu$), and the error bars extend one standard deviation ($\sigma$) above and below the mean.

### 3.3 Methods

We select representative methods of each motion group (Table 1) to focus on "basic" representations without too much focus on regularization tricks.

1. Low Order Poly+Fourier: **EffGS** (Katsumata et al., 2023) uses a 2-order fourier basis and a 1-order polynomial basis per Gaussian to model motion.
2. Low Order Poly+RBF: **SpaceTimeGaussians** (Li et al., 2023a) or **STG** uses a 3-order radial basis function and a 1-order polynomial basis per Gaussian to model motion. A small MLP decodes color; we remove this for fairer comparison (cf. **STG-decoder**).
3. Field MLP: **DeformableGS** (Yang et al., 2023) uses an MLP to estimate Gaussian offsets from a deformation field; the MLP is shared among all Gaussians.
4. Field HexPlane: **4DGS** (Wu et al., 2023) discretizes and factorizes the field with a HexPlane (Cao & Johnson, 2023); this is shared among all Gaussians.
5. 4D Gaussian: **RTGS** (Yang et al., 2024); this has no explicit motion model.

To minimize differences in implementation except for the motion model, we integrate these algorithms into a single codebase. For hyperparameters, we follow the original works closely, e.g., we use separate hyperparameters for synthetic and real-world scenes. Implementation details (e.g., motion-model degree, deformation-field depth, regularization loss weight, and learning rate) can be found in appendix C.

**Baselines:** We also evaluate **3DGS** without any motion, as adding a motion model might make results worse. We also include **TiNeuVox** (Fang et al., 2022) as a comparison point for what a voxel feature grid method can accomplish on dynamic scenes, even if it cannot be rendered quickly.

## 4 Results

We present our main findings and include full details in the supplemental material. To help frame our findings, we note that our implementations approximately match the results of the original works on their reported datasets. We also note that most dynamic Gaussian methods do not report performance across all datasets, and no dynamic Gaussian method has yet reported performance metrics on the iPhone dataset, so the comparisons made possible by this full-slate evaluation are of special interest.

### 4.1 Finding 1: Gaussian Methods Struggle In Comparison to a Hybrid Neural Field

We begin our analysis by evaluating all methods on all datasets. We present a summary of the findings, averaged across D-NeRF, Nerfies, HyperNeRF, NeRF-DS and iPhone datasets in Table 2. The full results are available in supplementary Section A.

Table 2: **Summary of Quantitative Results.** Table shows a summarized quantitative evaluation of all methods averaged across all five datasets.

| Method\Metric | PSNR↑ | SSIM↑ | MS-SSIM↑ | LPIPS↓ | FPS↑ | TrainTime (s)↓ |
|---|---|---|---|---|---|---|
| TiNeuVox | **24.54** | 0.706 | **0.804** | 0.349 | 0.29 | **2664.89** |
| 3DGS | 19.48 | 0.651 | 0.688 | 0.358 | **243.47** | 2964.81 |
| EffGS | 21.84 | 0.672 | 0.725 | 0.347 | 177.21 | 3757.81 |
| STG-decoder | 21.81 | 0.678 | 0.742 | 0.352 | 109.42 | 5980.64 |
| STG | 19.51 | 0.583 | 0.643 | 0.475 | 181.70 | 5359.56 |
| DeformableGS | 24.07 | 0.694 | 0.755 | 0.283 | 20.20 | 6227.43 |
| 4DGS | 23.55 | **0.708** | 0.765 | **0.277** | 62.99 | 8628.89 |
| RTGS | 21.61 | 0.663 | 0.720 | 0.350 | 143.37 | 7352.52 |

From this summary, we can make the broad and sobering observation that TiNeuVox, a non-Gaussian method, regularly outperforms all Gaussian methods in image quality. TiNeuVox also trains quickly and converges reliably, as compared to the Gaussian methods. The main drawback of TiNeuVox is simply its

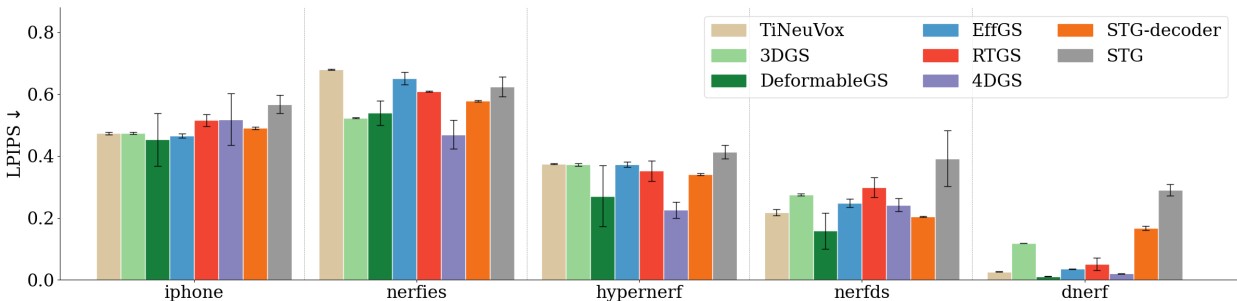

Figure 5: **Per-dataset Quantitative Results.** Test set LPIPS along with error bars for all methods on each of the different datasets. Note that lower is better. No method is clearly better across datasets.

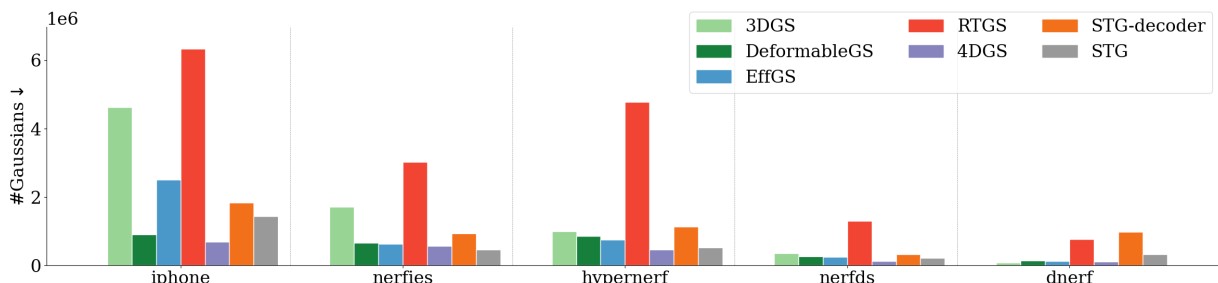

Figure 6: **# Gaussians** after optimization for each method, as a result of the adaptive densification.

rendering time, operating at 0.3 FPS compared to 20–200 FPS for the Gaussian methods. The speed difference is due to the rasterization in 3DGS versus volume rendering in NeRF.

### 4.2 Finding 2: Low-dimensional motion representations help

Comparing motion-based and 4D representations in Table 2, it appears that local, low-dimensional representations of motion perform better than less-constrained systems.

**Motion locality helps quality.** Comparing Gaussian methods on the LPIPS metric, we can observe that the field-based methods (DeformableGS & 4DGS) perform better than the methods which attach motions to individual Gaussians (EffGS & STGs) in rendering quality.

**Motion representation complexity hurts efficiency.** Comparing methods on training time and rendering speed, we find that basis-based methods (EffGS & STGs) are faster to train and faster to render than MLP-based methods (DeformableGS & 4DGS).

**Going to 4D makes things worse.** The 4D Gaussian approach, which is the most expressive representation of space-

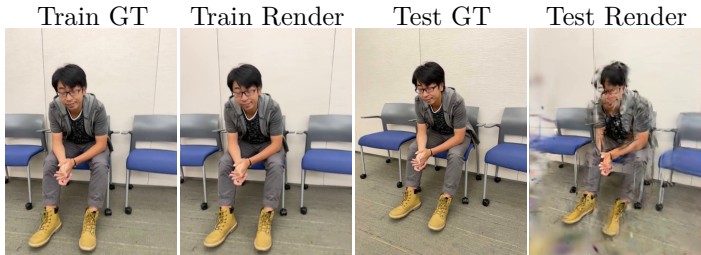

Train GT    Train Render    Test GT    Test Render

Figure 4: RTGS overfits to training views

time, performs worst in both quality and efficiency. This is consistent with our expectations established in Section 2.1. We qualitatively demonstrate this effect in Figure 4.

### 4.3 Finding 3: Dataset Variations Overwhelm Gaussian Method Variations

Inspecting the results of Gaussian methods *across* datasets, we find, contrary to the claims in the individual works, that a rank-ordering of the methods is unclear. We plot the per-dataset performance metrics in Figure 5, and note that the datasets have different winning methods. We observe that 4DGS and DeformableGS perform relatively well, but their performance across datasets varies greatly.

### 4.4 Finding 4: Adaptive Density Control Causes Headaches

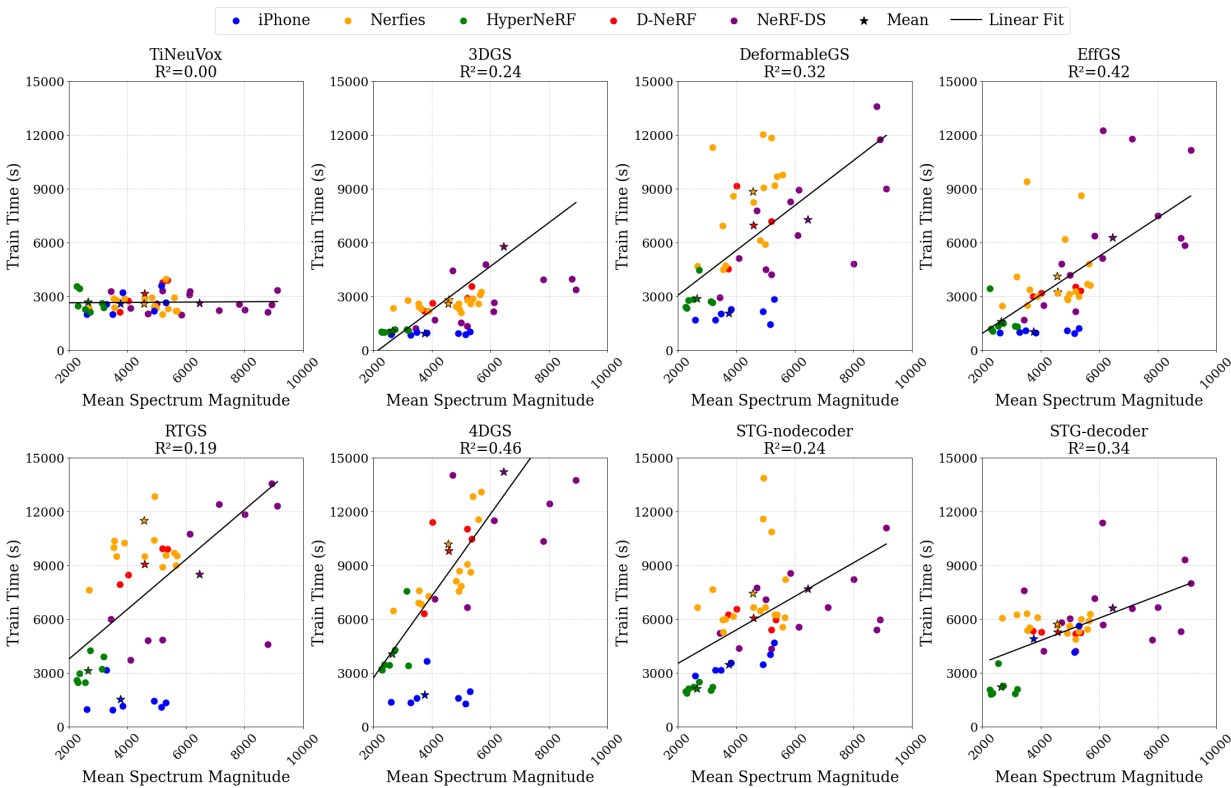

Figure 7: **Convergence vs. frequency.** We see that scenes with higher frequencies take longer to optimize.

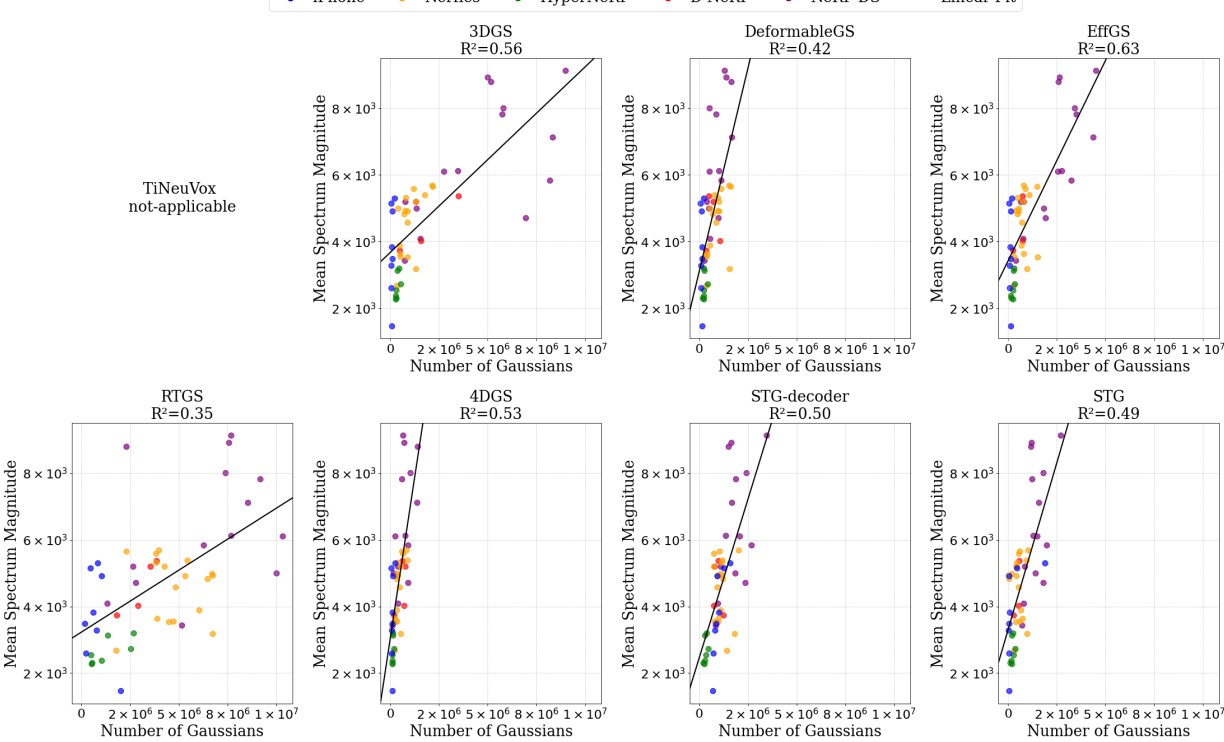

Figure 8: **Frequency vs. #Gaussians.** Scenes with higher frequencies end up with more Gaussians.

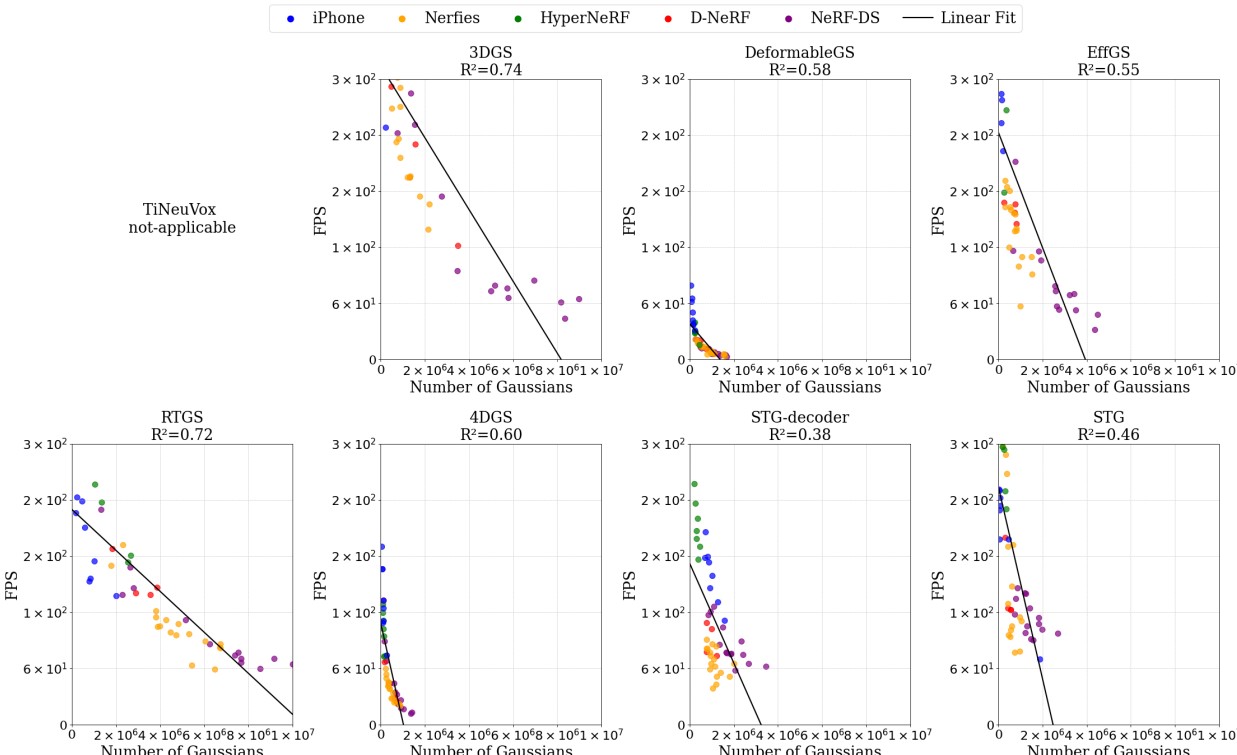

Figure 9: **Render FPS vs. #Gaussians.** More Gaussians leads to slower rendering.

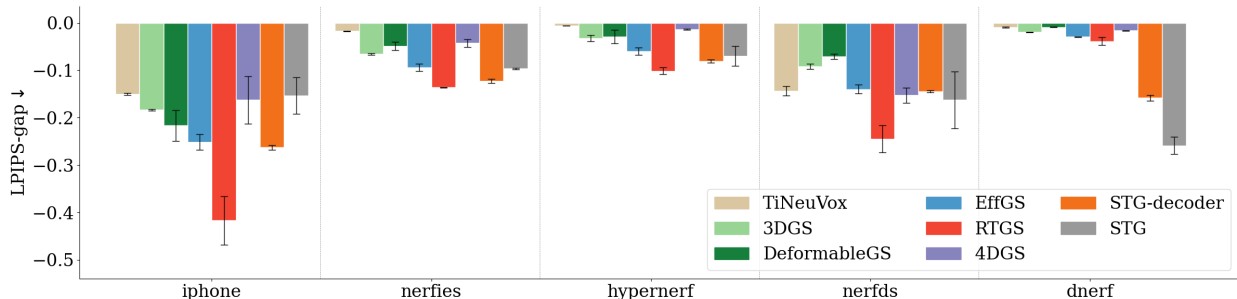

Figure 10: **Train-Test Performance Gaps.** We show the difference between the average LPIPS on the train and test set, where a larger gap indicates more overfitting to the training sequence. Note that here larger negative values indicate more severe overfitting.

As discussed in Section 2.1, a key detail of Gaussian Splatting methods is adaptive density control: Gaussians are added and removed during optimization, according to carefully tuned heuristics, allowing the complexity of the representation to adapt to the complexity of the data.

**Training and rendering efficiency varies greatly across scenes.** The optimization time for Gaussian Splatting methods varies greatly across scenes, with certain scenes consistently optimizing quickly and others optimizing slowly, across all methods.

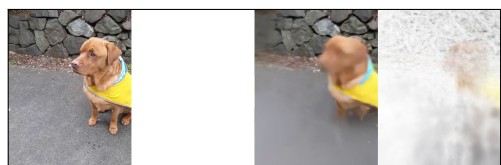

Figure 11: **Instability.** Training frame and renderings from 3 runs of the field MLP method DeformableGS (Yang et al., 2023).

Further, the methods take significantly more time to reconstruct real-world datasets (HyperNeRF, iPhone) compared to the synthetic datasets.

This factor is partially explained by the *frequency content* of the data. For this analysis, we use a Fast Fourier Transform (FFT) to transform each frame to the frequency domain, and aggregate the magnitude

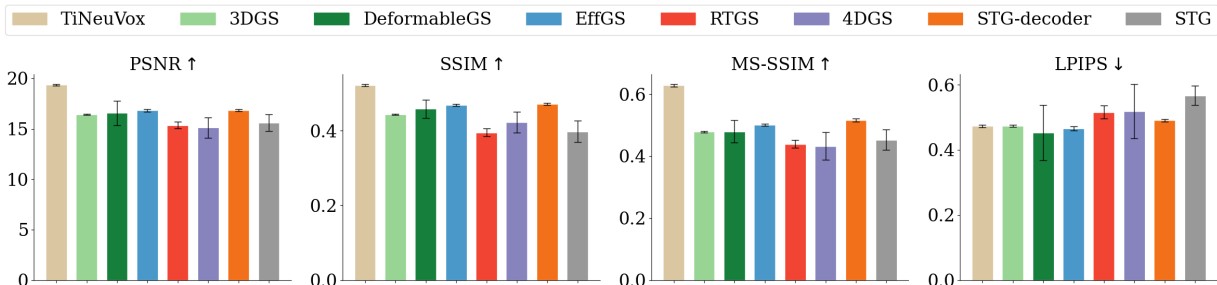

Figure 12: **Strictly-Monocular Dataset.** Test metrics on the strictly-monocular **iPhone** dataset.

spectrum from all images to calculate the mean frequency. Scenes with a predominance of higher frequencies take longer to optimize (Figure 7), because more Gaussian splats are required for reconstruction (Figure 8).

The total number of Gaussians after optimization varies greatly across scenes (Figure 6). This variability directly impacts efficiency, as rendering time is naturally tied to the number of Gaussians optimized for each scene (Figure 9). Methods based on neural deformation fields (DeformableGS and 4DGS) have a much higher render time per Gaussian due to the inference time of the underlying MLPs.

While this adaptive density feature adds expressivity to the overall representation, we find that it causes noticeable brittleness through three main effects: efficiency variations, increased overfitting risk, and occasional optimization failures. These factors may explain why some prior works tune the number of Gaussians and the adaptive density settings on a per-scene basis (Kerbl et al., 2023).

**Density adaptation makes overfitting worse.** In Figure 10, we show the train-test performance gap for all Dynamic Gaussian methods, along with 3DGS and TiNeuVox. For this evaluation, we calculate LPIPS metrics for train and test set of the same sequence, and then substract test set metric results from train set metrics. Larger quantities in this evaluation indicate wider train-test gaps, and serve as an indicator for overfitting. The evaluation reveals that the methods with adaptive density (i.e., all methods except TiNeuVox) have consistently wider train-test gaps. The only exception is in the NeRF-DS dataset, where we hypothesize that the reflective objects in that data cause issues for all methods, flattening the performance disparities between them.

**Density adaptation risks total failure.** Density changes can occasionally result in catastrophic failures with empty scene reconstructions (Figure 11). Gaussians may be pruned due to their opacities being under a hand-chosen threshold, and once a substantial number are deleted then it is difficult to recover the scene structure by subsequent cloning and splitting. We find that this happens unpredictably; we exclude these runs from our evaluation statistics.

### 4.5 Finding 5: Lack of Multi-View Cues Hurts Dynamic Gaussians More

Most datasets used for monocular dynamic view synthesis consist of data where there is a slow-moving scene captured by a rapidly-moving camera. This circumstance allows methods to leverage multi-view cues for optimization. DyCheck (Gao et al., 2022) introduced **iPhone** dataset to evaluate methods on "strictly-monocular" scenarios, meaning that the camera is moving more naturally. Figure 12 quantifies all methods' performance on this dataset. DyCheck showed that NeRF-based methods' performance degrades in the face of strict monocular setting, and we find that GS-based methods suffer here too. In fact, excepting for the perceptual metric **LPIPS**, all dynamic Gaussian methods perform worse than NeRF-like **TiNeuVox** method by a large margin. This is consistent with our finding on overfitting: since the Gaussian methods are more susceptible to overfitting, they are also more likely to do poorly with data lacking Multi-View Clues.

### 4.6 Finding 6: Narrow Baselines and Fast Objects Cause Error

Camera and object motion can both influence reconstruction performance considerably (Gao et al., 2022). This is related to the amount of multi-view information available to constrain optimization, but can be studied in detail with the help of our instructive synthetic dataset. Figure 14 (left) shows a high-level overview of the performance of all methods, across different camera baselines and different motion ranges. As the camera's baseline decreases, and/or as the object motion increases, reconstructions degrade. This result also holds for each method **individually** (see supplemental).

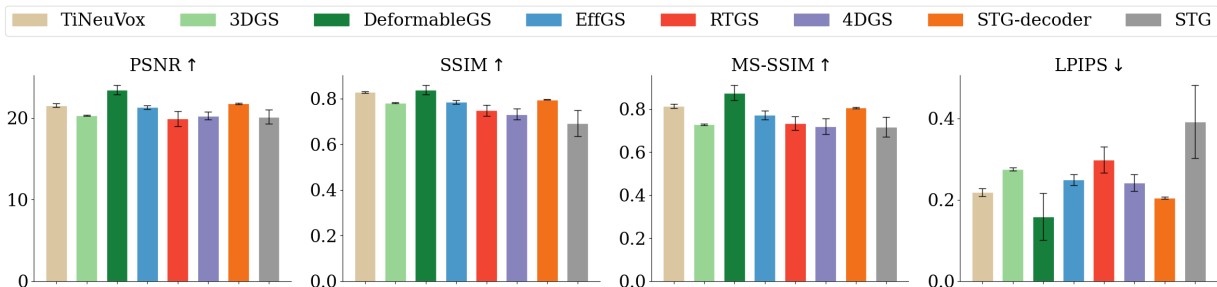

Figure 13: **Specular Dataset.** Test results on **NeRF-DS**, which includes reflective objects.

Figure 14 (right) summarizes LPIPS and masked LPIPS across methods and scenes in our instructive synthetic dataset. Unlike in the complex real-world datasets (Table 2), we see a clear ranking emerge between methods. DeformableGS is the most reliable and RTGS with EffGS are the least reliable. RTGS and STG occasionally crash during optimization, due to an out-of-memory error triggered as a side effect of the small-baseline setting. As such, we could not compute consistent metrics for STG in a narrow baseline setting (camera baseline $\leq 5$), and so we show two numbers in the figure: the metric computed over all baselines, where crashed runs are assigned minimum possible value for that metric (solid bar), and the metric computed over wide baselines only (textured bar). TiNeuVox is competitive in accordance with Section 4.1.

Curiously, 3DGS—a completely static method—achieves a high rank when using LPIPS over the whole image. This happens for several reasons. First, a large portion of the image is covered by static background, which the method is well-suited to reconstruct. Second, 3DGS can overfit the dynamic object to some extent by creating incorrect geometry on its motion path and making it appear when needed. The artifacts on the dynamic object are still pronounced, especially when the cube is rotating. In the masked LPIPS ranking 3DGS falls to the low end of the ranking as expected. In sum, we find that all methods are sensitive to camera baseline and the magnitude of the underlying motion, although the levels of robustness vary.

### 4.7 Finding 7: Specular Objects are a Challenge for All Methods

Yang et al. (2023) highlighted that their **DeformableGS** method outpeforms previous state-of-art methods on **NeRF-DS** dataset, including **TiNeuVox**. As NeRF-DS focuses on specular objects, this might imply that other Gaussian methods are able to handle specular objects well. However, we found that this is not the case (Figure 13). Generally, it is difficult for methods to distinguish between specular effects and small object motions especially under small baselines.

### 4.8 Finding 8: Foreground/Background Separation Clarifies Static/Dynamic Results

Inspecting the 3DGS performance in Table 2, we note that it performs surprisingly well, considering that it is a static scene representation. This is partly because the moving foreground constitutes a small fraction of the pixels in the given dataset sequences. We also find that 3DGS is able to smuggle a pseudo-dynamic scene into its representation, by inserting Gaussians which only render from certain viewpoints (and therefore only certain timesteps), and thereby reduce loss on dynamic parts of the training images. Please see the supplementary material for visualizations of this effect.

To better evaluate Dynamic Gaussians' advantage over 3DGS, we evaluate mLPIPS, mPSNR, mSSIM and mMS-SSIM (Figure 15). Comparing Figure 5 and Figure 15, we see that Dynamic Gaussians' performance does not change much after masking, but 3DGS's performance worsens dramatically, suggesting that the dynamic scene representations are indeed better at capturing the moving part of the scene.

## 5 Discussion on Motion Representations and Reconstruction Quality

Extending static 3D Gaussian Splatting (3DGS) to handle dynamic scenes from a monocular camera requires carefully choosing how to represent time and motion. Our experiments (Table 2 and Figure 5) show that the parameterization of motion can significantly affect both reconstruction quality and robustness—yet no single representation is universally superior (Finding 3). In this section, we summarize how these different representations influence performance, why certain approaches may excel in particular scenarios, and how scene complexity factors like multi-view coverage can overshadow the differences between methods.

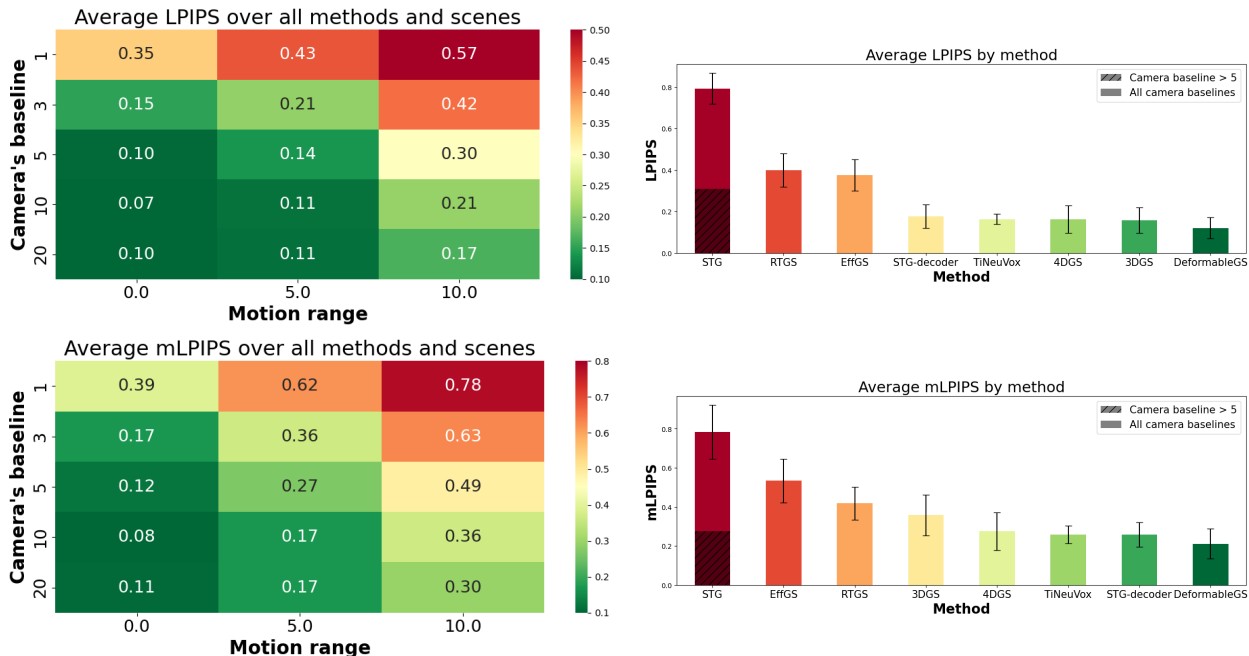

Figure 14: **Results on our instructive synthetic dataset.** *Top Left*: LPIPS↓ heatmap shows how camera baseline and object motion range affect performance across all methods on average. Decreasing object motion range (right to left) affects reconstruction performance positively; decreasing camera baseline (bottom to top) has the same effect. Values over 0.3 usually represent a failure to reconstruct the dynamic object. *Top Right*: Ranking of methods by average LPIPS over all scenes. Solid bars represent score computed over all scenes. Textured bar represents metrics computed on wide camera baselines. *Bottom Left*: Masked LPIPS↓ heatmap: with the focus on dynamic object reconstruction, the metrics become worse on average. *Bottom Right*: Ranking for masked LPIPS.

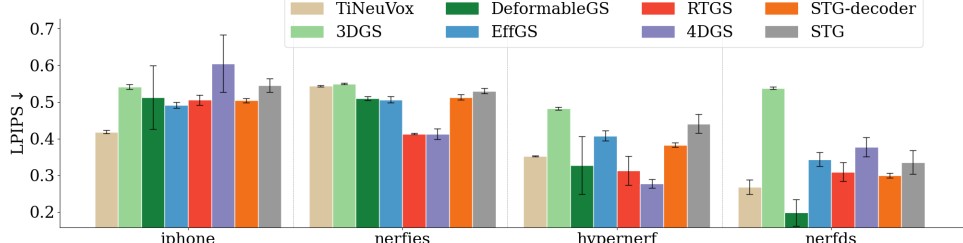

Figure 15: **Foreground-Only LPIPS (↓)** reveals the weakness of 3DGS on dynamic elements.

## 5.1 Why Motion Representations Matter

From an optimization standpoint, any dynamic splatting algorithm must determine *where* and *how* Gaussians move as the scene evolves over time. The chosen motion representation fundamentally affects:

- **Expressivity:** Does the method capture simple object translations only, or is it flexible enough for highly non-rigid deformations?

- **Robustness:** Under strictly-monocular settings (e.g., iPhone dataset), can it avoid overfitting (spurious geometry) or collapsing solutions?

- **Efficiency:** How does the motion model affect memory use, training speed, or runtime performance?

Even small changes in parameterization (e.g., low-order polynomials vs. MLP-based fields vs. 4D embeddings) can cause large performance variations, especially under challenging setups with limited camera baselines or significant scene/foreground motion (e.g., see Finding 6 and Figure 14).

## 5.2 Per-Gaussian 3D Motion vs. Field-Based 3D Motion vs. 4D Representations

**Low-Dimensional 3D.** One main distinction is whether each Gaussian carries its own motion parameters (e.g., low-order polynomials in EffGS (Katsumata et al. (2023)), STG (Li et al. (2023a))) or whether a *shared* deformation field provides offsets (DeformableGS (Yang et al. (2023)), 4DGS (Wu et al. (2023))). Per-Gaussian approaches are highly flexible, since each splat can move independently. However, this flexibility can also encourage overfitting (Finding 5, Figure 10 and Figure 12), particularly if the camera only sees each Gaussian from limited span of views. By contrast, field-based methods encourage spatial coherence among Gaussians—potentially boosting fidelity in dynamic regions, but at the cost of higher complexity and slower inference (Table 2, Figure 9, Figure 14, Figure 15).

**4D Gaussians (RTGS).** An alternative paradigm is to embed space and time directly in $\mathbb{R}^4$, removing the need for any explicit motion function. RTGS (Yang et al. (2024)), for example, defines each Gaussian in a unified 4D volume. While this representation is appealing for complex deformations, Finding 3 and Figure 4 show that it can overfit severely in narrow-baseline or fast-motion scenarios if multi-view cues are lacking (also see Finding 5). Rather than guaranteeing better fidelity, the added degrees of freedom can produce flickering or distorted geometry when the data is insufficient to anchor Gaussians in time.

## 5.3 Overfitting and Adaptive Density Control

All dynamic Gaussian splatting methods rely on *adaptive density control* to split, clone, or prune Gaussians over the course of training. While this enhances expressivity (capturing fine details as the scene changes), it also heightens the risk of overfitting. Figure 10 shows how these methods tend to have a larger train–test gap than simpler baselines like TiNeuVox (see Finding 1). In addition, adaptive density can lead to unpredictable optimization failures (Finding 4, Figure 11). Tuning this mechanism thus remains a critical factor for successful reconstructions.

## 5.4 Dataset-Specific Factors Often Dominate

Despite theoretical differences among motion models, scene complexity and dataset factors often override these distinctions:

- **Strictly-Monocular Sequences:** iPhone data (see Figure 12) with fast-moving objects and smaller camera baselines reveal bigger gaps in performance; most dynamic Gaussian methods lag behind a voxel-based approach (TiNeuVox) that is slower to render but less prone to overfitting (see Finding 5).

- **High-Frequency or Reflective Surfaces:** Scenes contain visual cues that mimic motion can confuse any parameterization (as in NeRF-DS, Figure 13).

- **Fast Foreground Motion:** Our synthetic experiments (Finding 6, Figure 14, Figure 15) show how narrow camera baselines plus large object motion are especially prone to reconstruction failures.

Thus, *which* motion model you choose is only part of the story; the constraints provided by the scene (e.g., multi-view coverage, stable lighting, fewer specularities) can overshadow method-centric differences.

## 5.5 Guidelines and Implications

Taken together, our findings suggest:

- **Match Representation to Motion Demands.** Field-based (shared) motion can capture non-rigid deformations more effectively, but might be computationally heavier (Finding 2, Table 2). Per-Gaussian polynomial or Fourier motion is simpler to train yet risks localized overfitting if the camera baseline is too small.

- **Simplify When Data Are Limited.** In challenging scenarios (e.g., narrow baselines, specular objects), strongly constraining motion often prevents degenerate solutions (Finding 5).

- **Use Caution with Adaptive Density Control.** Splitting and pruning Gaussians can add detail but also destabilize optimization (Finding 4).

- **Consider Foreground vs. Background Metrics.** Masked metrics (Finding 8) reveal a more accurate picture of dynamic-object quality; a static 3DGS approach may score well on overall PSNR if most pixels are stationary, even though it fails in regions with actual motion.

Ultimately, our evaluations suggest that *monocular dynamic gaussian splatting is fast and brittle, and scene complexity rules.* In other words, no single motion representation is universally ideal. Instead, the "best" approach is highly context-dependent, shaped by the complexity of the scene, the camera trajectory, and the interplay of adaptive density updates. Even so, awareness of each representation's inherent trade-offs can guide practitioners in selecting or developing dynamic Gaussian splatting methods that balance robustness, expressivity, and computational cost in real-world settings.

## 6 Conclusion

With the emerging popularity of Gaussian splatting, multiple works were recently introduced simultaneously that tackle the challenging setting of dynamic scene view synthesis using only monocular input, each claiming superior performance even with only minor methodological differences and inconsistent evaluation settings. We organize, consolidate, benchmark, and analyze many of these Gaussian splatting-based methods and provide a shared, consistent, apples-to-apples comparison between them on instructive synthetic data and complex real-world data. We also define conceptual differences between these methods and analyze their impact on a variety of performance metrics. This work may lead to further progress in dynamic GS methods, with broader impact in areas such as video editing, visual art, and 3D scene analysis for industrial applications.

**Limitations.** Evaluations like ours would benefit from being able to describe each sequence's reconstruction complexity independently of any method's performance upon that sequence. With this, it would be simpler to find insights by correlating methods with scene difficulty. Establishing such a metric is a chicken-and-egg problem: it requires measuring the geometry and motion of a dynamic scene, which is our initial problem. Past works have proposed approximations to a scene's difficulty, but so far these metrics do not explain reconstruction quality well, e.g., DyCheck's (Gao et al., 2022) $\omega$ requires only camera pose but does not correlate to reconstruction quality (Figure 16). Similarly, ground-truth evaluation of reconstruction such as for depth or motion would also aid in this task; however, capturing dense ground truth is difficult for real-world dynamic scenes (e.g., DyCheck's $\Omega$ requires sparse ground truth points and so cannot be widely used). Next, in this rapidly-growing field, we could not include all concurrent works in our analysis; our publicly-released code and data will make it possible to supplement our analysis with additional works. Finally, we note that various regularization strategies have been proposed to enhance stability and resolve ambiguities. We regard these as promising directions for future exploration and provide a brief survey in Appendix F.

**Acknowledgements.** YL, MO, RL, JT thank NSF CAREER 2144956 and NASA RI-80NSSC23M0075.

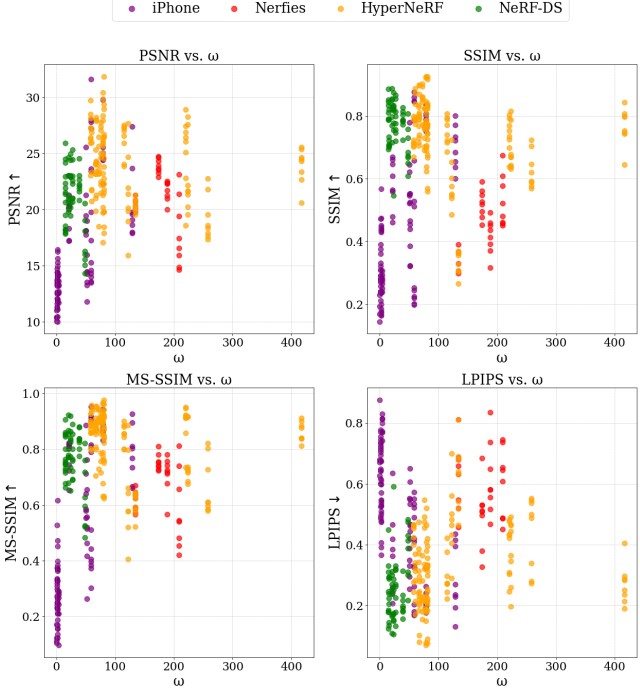

Figure 16: **DyCheck's $\omega$ sequence difficulty metric vs. PSNR↑/SSIM↑/MS-SSIM↑/LPIPS↓.** Each dot is a sequence from a dataset. There is no strong correlation either within or across datasets.

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

# A   Additional Results

Please check the webpage for qualitative results on both existing datasets and our instructive dataset.

## A.1   Quantitative Results Per Dataset

Table 3: **Summary of Quantitative Results.** Table shows a summarized quantitative evaluation of all methods averaged across D-NeRF dataset.

| Method\Metric | PSNR↑ | SSIM↑ | MS-SSIM↑ | LPIPS↓ | FPS↑ | TrainTime (s)↓ |
|---|---|---|---|---|---|---|
| TiNeuVox | 33.03 | 0.97 | 0.99 | 0.03 | 0.42 | 2582.54 |
| 3DGS | 20.64 | 0.92 | 0.88 | 0.12 | **340.85** | **928.83** |
| EffGS | 30.52 | 0.96 | 0.98 | 0.04 | 289.47 | 1042.04 |
| STG-decoder | 25.89 | 0.91 | 0.90 | 0.17 | 160.32 | 3462.29 |
| STG | 17.09 | 0.88 | 0.66 | 0.29 | 208.98 | 4889.50 |
| DeformableGS | **37.14** | **0.99** | **0.99** | **0.01** | 50.78 | 2048.38 |
| 4DGS | 33.27 | 0.98 | 0.99 | 0.02 | 134.13 | 1781.46 |
| RTGS | 28.78 | 0.96 | 0.96 | 0.05 | 192.37 | 1519.60 |

Table 4: **Summary of Quantitative Results.** Table shows a summarized quantitative evaluation of all methods averaged across HyperNeRF dataset.

| Method\Metric | PSNR↑ | SSIM↑ | MS-SSIM↑ | LPIPS↓ | FPS↑ | TrainTime (s)↓ |
|---|---|---|---|---|---|---|
| TiNeuVox | **26.45** | 0.74 | 0.88 | 0.38 | 0.12 | 2604.45 |
| 3DGS | 20.98 | 0.69 | 0.76 | 0.37 | **244.87** | **2587.02** |
| EffGS | 22.24 | 0.70 | 0.79 | 0.37 | 138.79 | 4119.66 |
| STG-decoder | 23.92 | 0.73 | 0.83 | 0.34 | 66.72 | 7423.41 |
| STG | 22.92 | 0.70 | 0.78 | 0.41 | 183.21 | 5729.80 |
| DeformableGS | 24.58 | 0.74 | 0.83 | 0.27 | 10.91 | 8855.46 |
| 4DGS | 25.70 | **0.79** | **0.89** | **0.23** | 37.46 | 10170.86 |
| RTGS | 22.99 | 0.71 | 0.79 | 0.35 | 104.64 | 11507.80 |

Table 5: **Summary of Quantitative Results.** Table shows a summarized quantitative evaluation of all methods averaged across NeRF-DS dataset.

| Method\Metric | PSNR↑ | SSIM↑ | MS-SSIM↑ | LPIPS↓ | FPS↑ | TrainTime (s)↓ |
|---|---|---|---|---|---|---|
| TiNeuVox | 21.54 | 0.83 | 0.81 | 0.22 | 0.49 | 2696.48 |
| 3DGS | 20.29 | 0.78 | 0.73 | 0.28 | **353.99** | **1066.48** |
| EffGS | 21.28 | 0.78 | 0.77 | 0.25 | 307.70 | 1597.90 |
| STG-decoder | 21.73 | 0.80 | 0.81 | 0.20 | 212.24 | 2131.48 |
| STG | 20.13 | 0.69 | 0.72 | 0.39 | 302.89 | 2214.62 |
| DeformableGS | **23.42** | **0.84** | **0.88** | **0.16** | 30.27 | 2885.07 |
| 4DGS | 20.25 | 0.73 | 0.72 | 0.24 | 100.22 | 4075.43 |
| RTGS | 19.88 | 0.75 | 0.73 | 0.30 | 259.83 | 3116.21 |

Table 6: **Summary of Quantitative Results.** Table shows a summarized quantitative evaluation of all methods averaged across Nerfies dataset.

| Method\Metric | PSNR↑ | SSIM↑ | MS-SSIM↑ | LPIPS↓ | FPS↑ | TrainTime (s)↓ |
|---|---|---|---|---|---|---|
| TiNeuVox | **22.89** | 0.45 | 0.71 | 0.68 | 0.11 | 3145.67 |
| 3DGS | 20.14 | 0.46 | 0.66 | 0.52 | **209.80** | **2823.33** |
| EffGS | 20.13 | 0.43 | 0.61 | 0.65 | 159.17 | 3249.33 |
| STG-decoder | 20.93 | 0.47 | 0.67 | 0.58 | 90.46 | 6050.42 |
| STG | 20.55 | 0.46 | 0.66 | 0.62 | 142.30 | 5264.58 |
| DeformableGS | 21.26 | 0.43 | 0.66 | 0.54 | 14.99 | 6950.94 |
| 4DGS | 21.84 | **0.50** | **0.72** | **0.47** | 35.70 | 9797.50 |
| RTGS | 20.06 | 0.42 | 0.62 | 0.61 | 153.38 | 9059.54 |

Table 7: **Summary of Quantitative Results.** Table shows a summarized quantitative evaluation of all methods averaged across iPhone dataset.

| Method\Metric | PSNR↑ | SSIM↑ | MS-SSIM↑ | LPIPS↓ | FPS↑ | TrainTime (s)↓ |
|---|---|---|---|---|---|---|
| TiNeuVox | **19.35** | **0.52** | **0.63** | 0.47 | 0.36 | **2632.17** |
| 3DGS | 16.41 | 0.44 | 0.48 | 0.47 | **140.50** | 5777.45 |
| EffGS | 16.82 | 0.47 | 0.50 | 0.47 | 99.64 | 6275.33 |
| STG-decoder | 16.85 | 0.47 | 0.52 | 0.49 | 86.18 | 7694.83 |
| STG | 15.60 | 0.40 | 0.45 | 0.57 | 114.95 | 6629.64 |
| DeformableGS | 16.56 | 0.46 | 0.48 | **0.45** | 10.47 | 7278.28 |
| 4DGS | 15.13 | 0.42 | 0.43 | 0.52 | 42.53 | 14205.48 |
| RTGS | 15.36 | 0.39 | 0.44 | 0.52 | 101.33 | 8484.05 |

## A.2 Quantitative Result Plots Per Metric

### A.2.1 PSNR

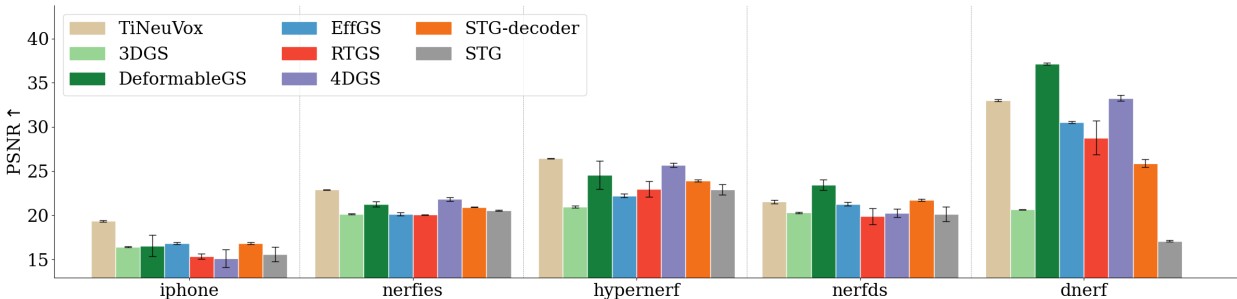

Figure 17: **Per-dataset Quantitative Results.** Test set PSNR along with error bars for all methods on each of the datasets. (↑). We see that the datasets have different winning methods.

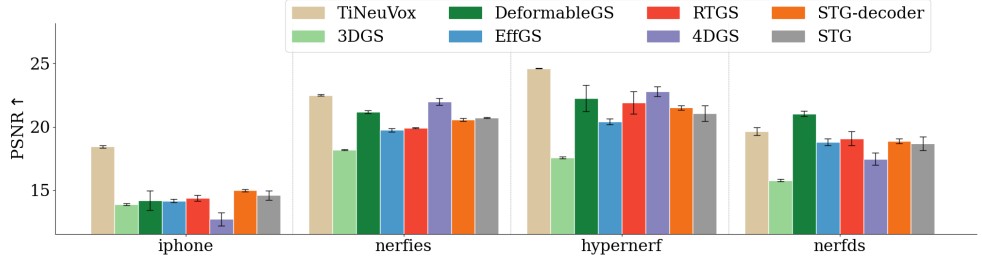

Figure 18: **Foreground-Only PSNRs Evaluation (↑).**

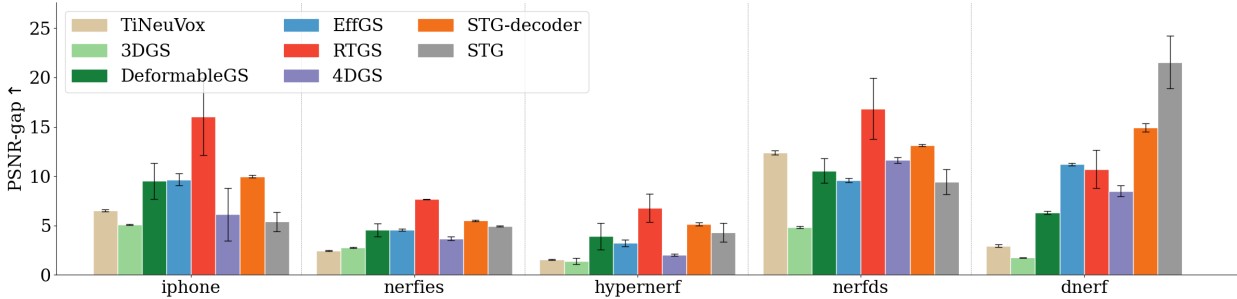

Figure 19: **Train-Test Performance Gaps.** We show the difference between the average PSNR on the train and test set, where a larger gap indicates more overfitting to the training sequence.

### A.2.2 SSIM

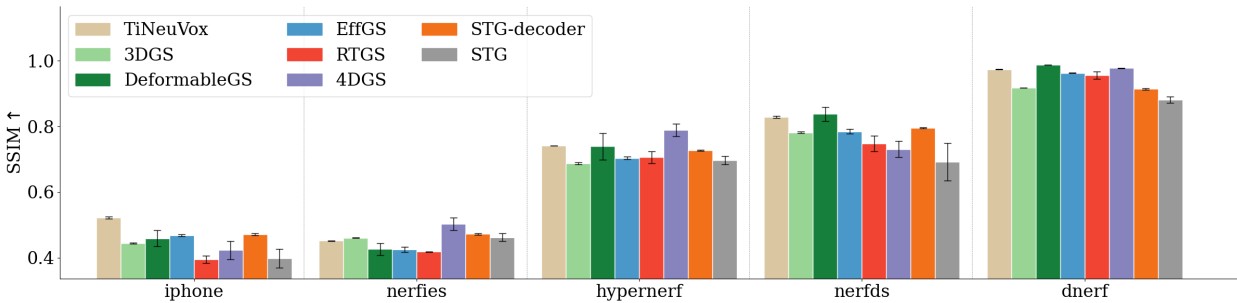

Figure 20: **Per-dataset Quantitative Results.** Test set SSIM along with error bars for all methods on each of the datasets. (↑). We see that the datasets have different winning methods.

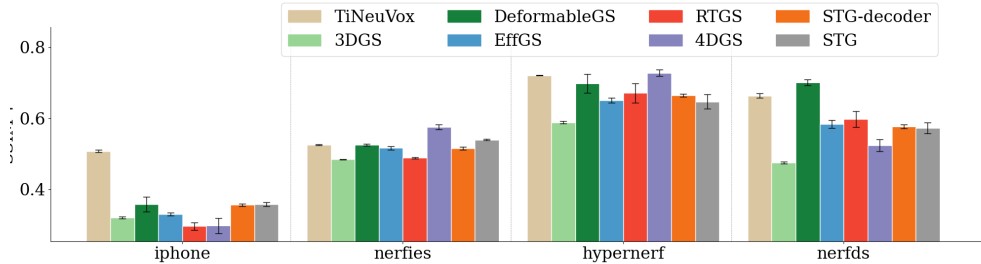

Figure 21: **Foreground-Only SSIMs Evaluation (↑).**

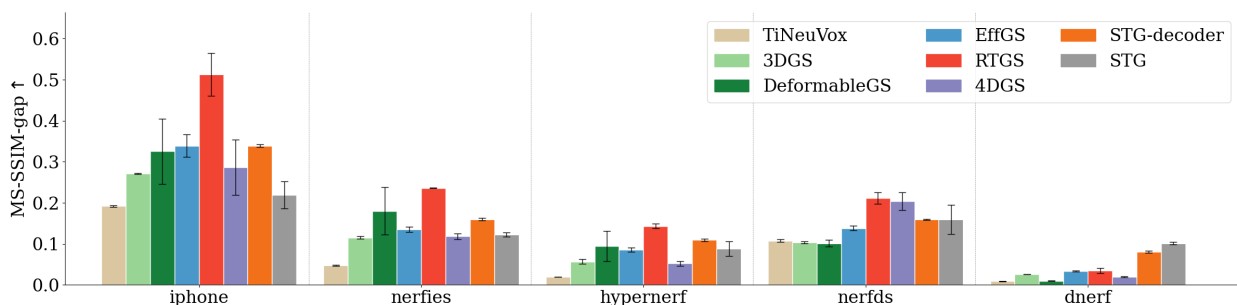

Figure 22: **Train-Test Performance Gaps.** We show the difference between the average SSIM on the train and test set, where a larger gap indicates more overfitting to the training sequence.

### A.2.3 MS-SSIM

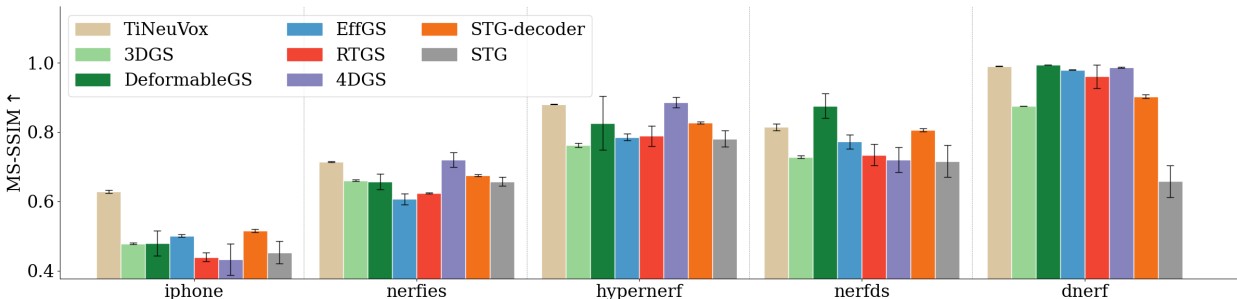

Figure 23: **Per-dataset Quantitative Results.** Figure shows the test set MS-SSIM along with error bars for all methods on each of the different datasets. Note that higher is better. We see that the datasets have different winning methods.

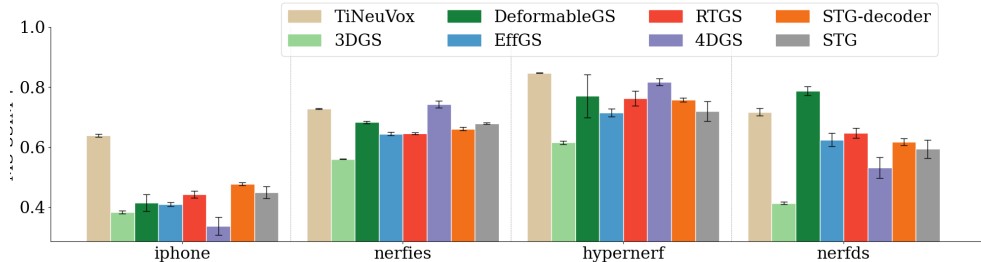

Figure 24: **Foreground-Only MS-SSIMs Evaluation (↑).**

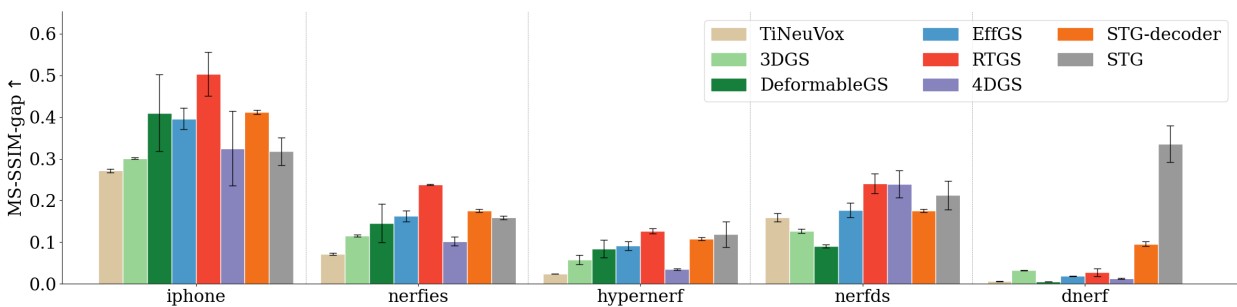

Figure 25: **Train-Test Performance Gaps.** We show the difference between the average MS-SSIM on the train and test set, where a larger gap indicates more overfitting to the training sequence.

### A.2.4   Train Time and FPS

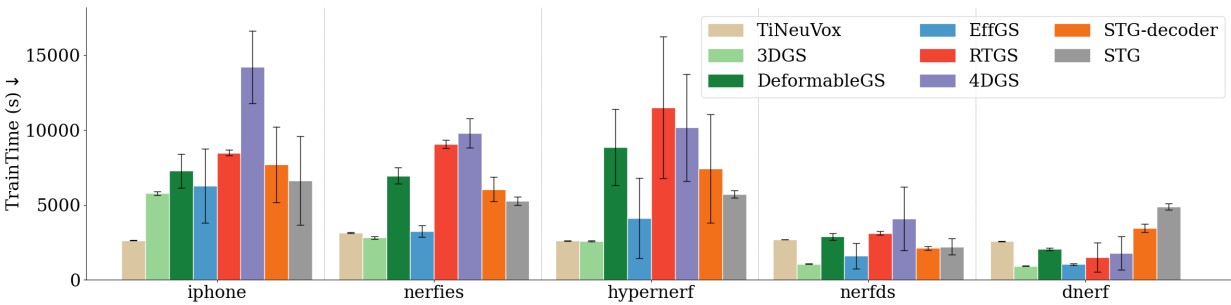

Figure 26: **Per-dataset Quantitative Results.** Figure shows the test set Train Time along with error bars for all methods on each of the different datasets. Note that lower is better. We see that the datasets have different winning methods.

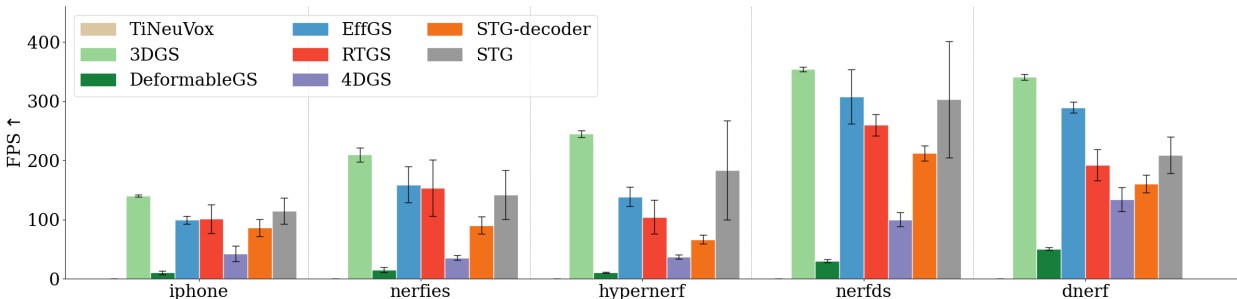

Figure 27: **Per-dataset Quantitative Results.** Figure shows the test set render FPS along with error bars for all methods on each of the different datasets. Note that higher is better. We see that the datasets have different winning methods.

### A.2.5   Relative Gap vs. Absolute Performance on NeRF-DS dataset

In section 4.4, we noted that reflective objects cause consistent difficulties for all methods, thereby "flattening" performance disparities. By "flattening," we were referring to the *relative* gaps among the methods, rather than an *absolute* drop in metrics. Because each method struggles similarly with specularities, the difference in performance between methods shrinks, and no single approach outperforms the others by a large margin.

However, this Appendix section shows that, in an absolute sense, average-quality measures (PSNR, SSIM, etc.) on NeRF-DS do not degrade as dramatically as one might expect. We attribute this to two factors:

1. **Static Background Influence.** Like many datasets, NeRF-DS contains substantial non-reflective or relatively static image regions. Even with reflections, each method can still capture those simpler background regions well enough that the scene-level averages do not plummet.

2. **Global Averaging Subtle Artifacts.** Reflective artifacts often occur in smaller foreground regions. While these artifacts degrade local image quality, they may have a limited effect on global metrics averaged over entire test frames. As a result, the overall PSNR/SSIM may remain comparable to those on other datasets.

In short, the "flattening" reflects how the presence of specularities hinders every method similarly, reducing the relative performance spreads. Yet, on the absolute scale, large swathes of non-reflective or simpler regions help keep dataset-level averages from showing a dramatic overall performance drop.

### A.3 Ablations for Instructive Dataset

Figures 28, 29, 31 and 33 contain ablation heatmaps for PSNR, SSIM, MS-SSIM, and their masked variants. Figures 30, 32 and 34 show the aggregate metrics and rankings. The rankings are similar to those in fig. 14. We note that 3DGS often ends up higher in the ranking for pixel-difference metrics mPSNR, mSSIM, mMS-SSIM, than mLPIPS likely due to averaging with strong performance on the static part of the dataset (motion range = 0) and pixel-difference metrics' propensity to overfit.

#### A.3.1 LPIPS and mLPIPS

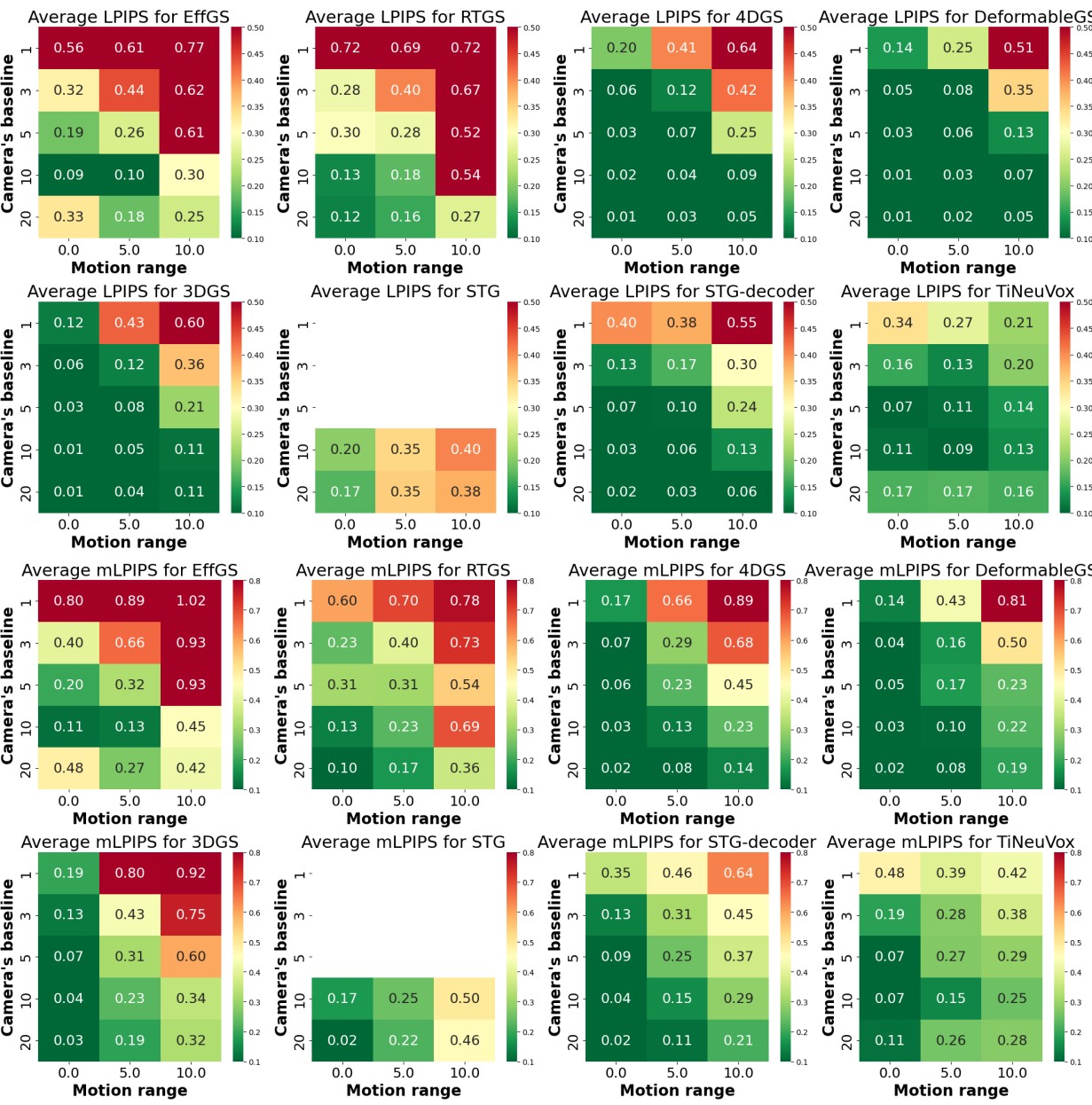

Figure 28: **Individual ablations for the instructive dataset, measured by LPIPS.** The figure shows camera baseline and motion range ablations for each method separately. *Top Two Rows:* LPIPS↓ heatmaps. *Bottom Two Rows:* mLPIPS↓ heatmaps. On average the reconstruction becomes harder with increase of the motion range (left to right) and with decrease of the camera baseline (bottom to top). Missing values indicate a non-converging model.

## A.3.2 PSNR and mPSNR

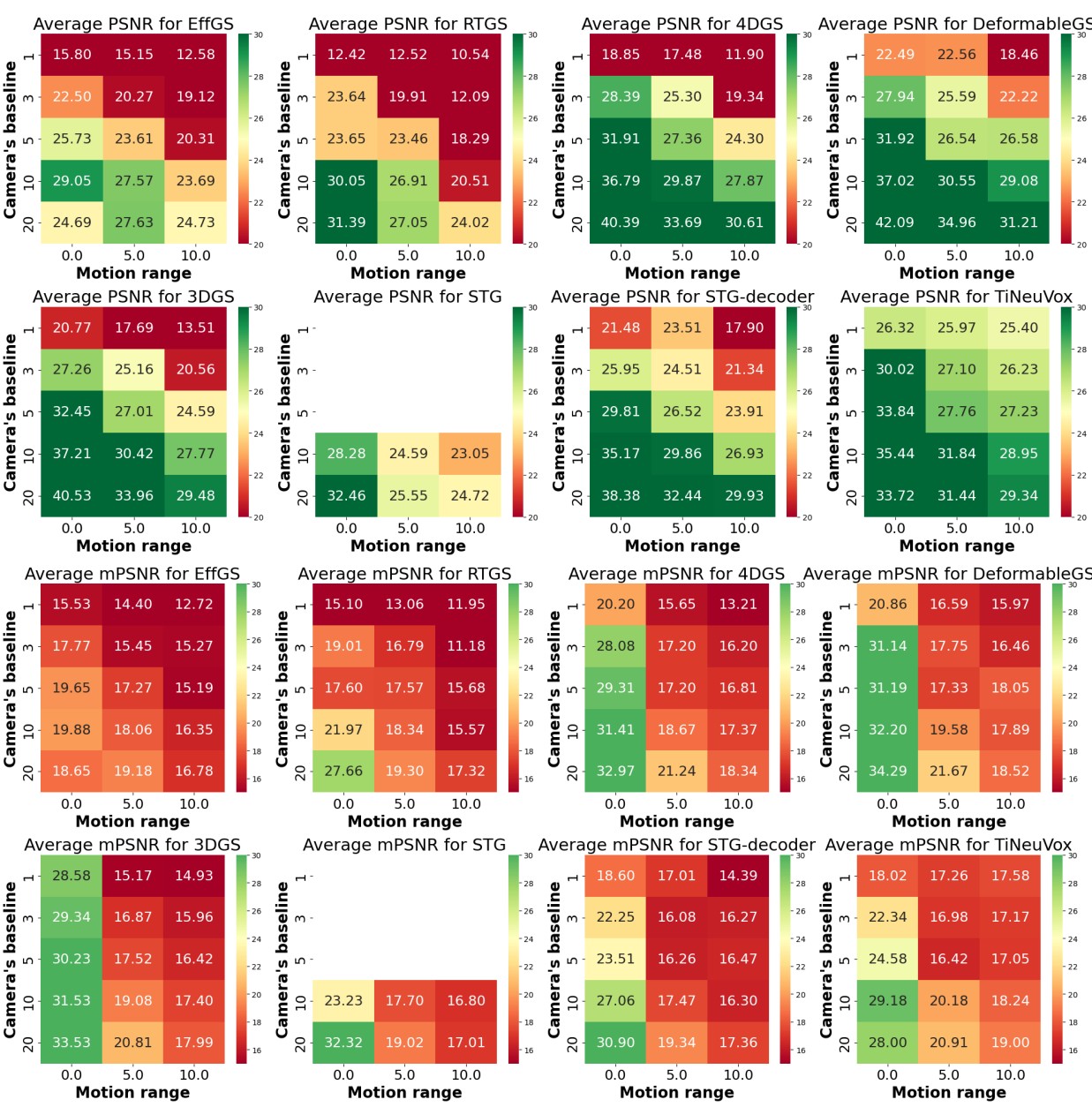

Figure 29: **Individual ablations for the instructive dataset, measured by PSNR.** The figure shows camera baseline and motion range ablations for each method separately. *Top Two Rows:* PSNR↑ heatmaps. *Bottom Two Rows:* mPSNR↑ heatmaps. On average the reconstruction becomes harder with increase of the motion range (left to right) and with decrease of the camera baseline (bottom to top). Missing values indicate a non-converging model.

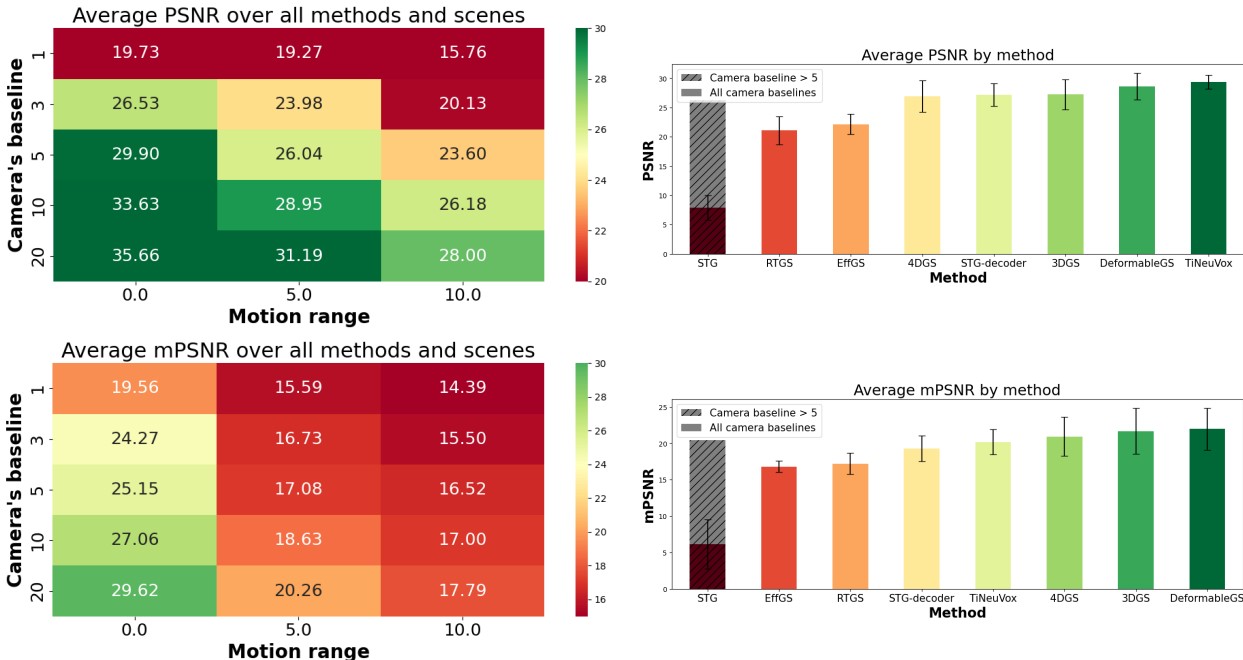

Figure 30: **Results on our instructive synthetic dataset.** *Top Left*: PSNR↑ heatmap shows how camera baseline and object motion range affect performance across all methods on average. Increasing object motion range (left to right) affects reconstruction performance negatively; decreasing camera baseline (bottom to top) has the same effect. *Top Right*: Ranking of methods by average PSNR over all scenes. Solid bars represent score computed over all scenes. Textured bar represents metrics computed on wide camera baselines. *Bottom Left*: Masked PSNR↑ heatmap: with the focus on dynamic object reconstruction, the metrics become worse on average. *Bottom Right*: Ranking for masked PSNR.

### A.3.3    SSIM and mSSIM

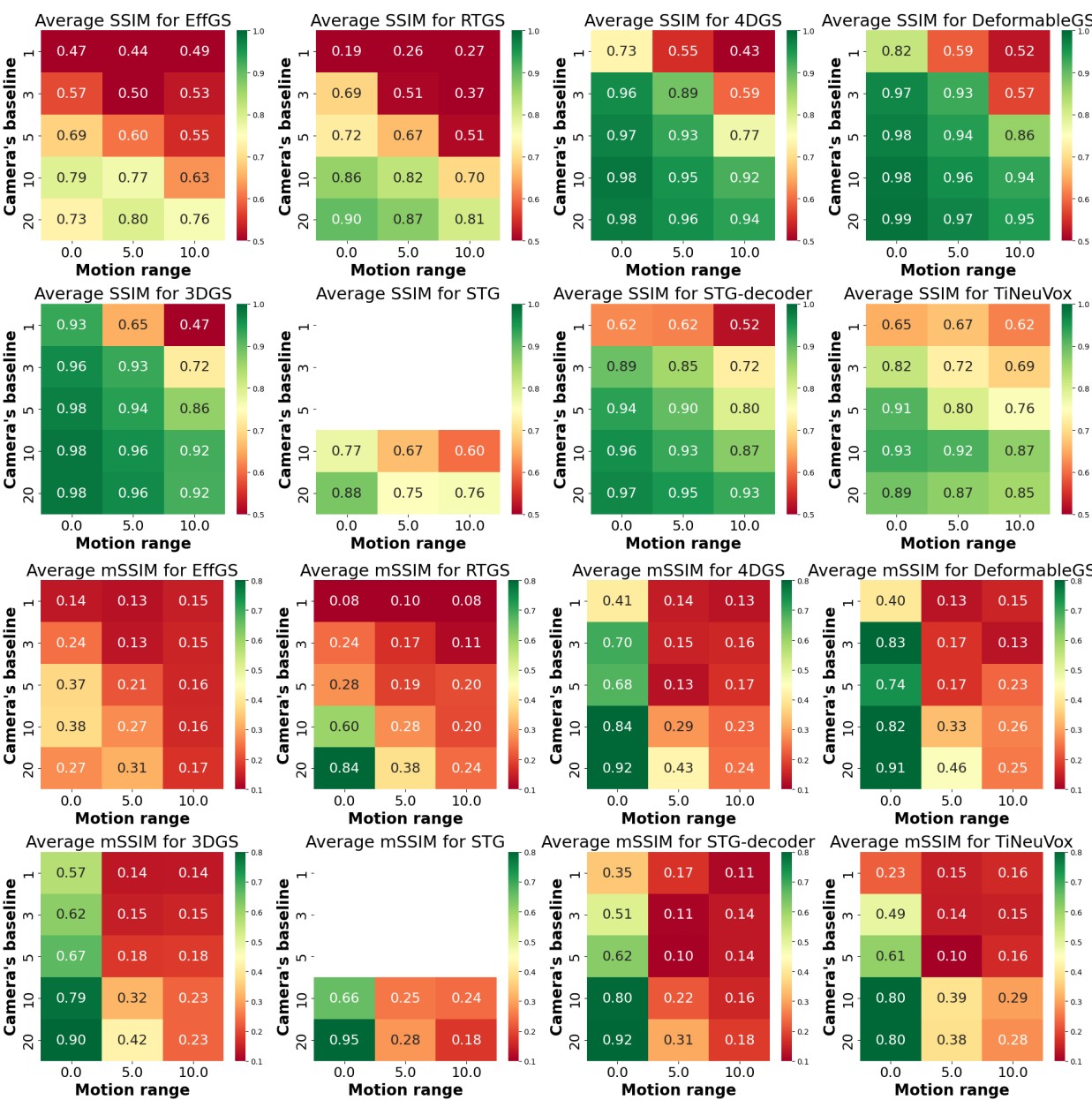

Figure 31: **Individual ablations for the instructive dataset, measured by SSIM.** The figure shows camera baseline and motion range ablations for each method separately. *Top Two Rows:* SSIM↑ heatmaps. *Bottom Two Rows:* mSSIM↑ heatmaps. On average the reconstruction becomes harder with increase of the motion range (left to right) and with decrease of the camera baseline (bottom to top). Missing values indicate a non-converging model.

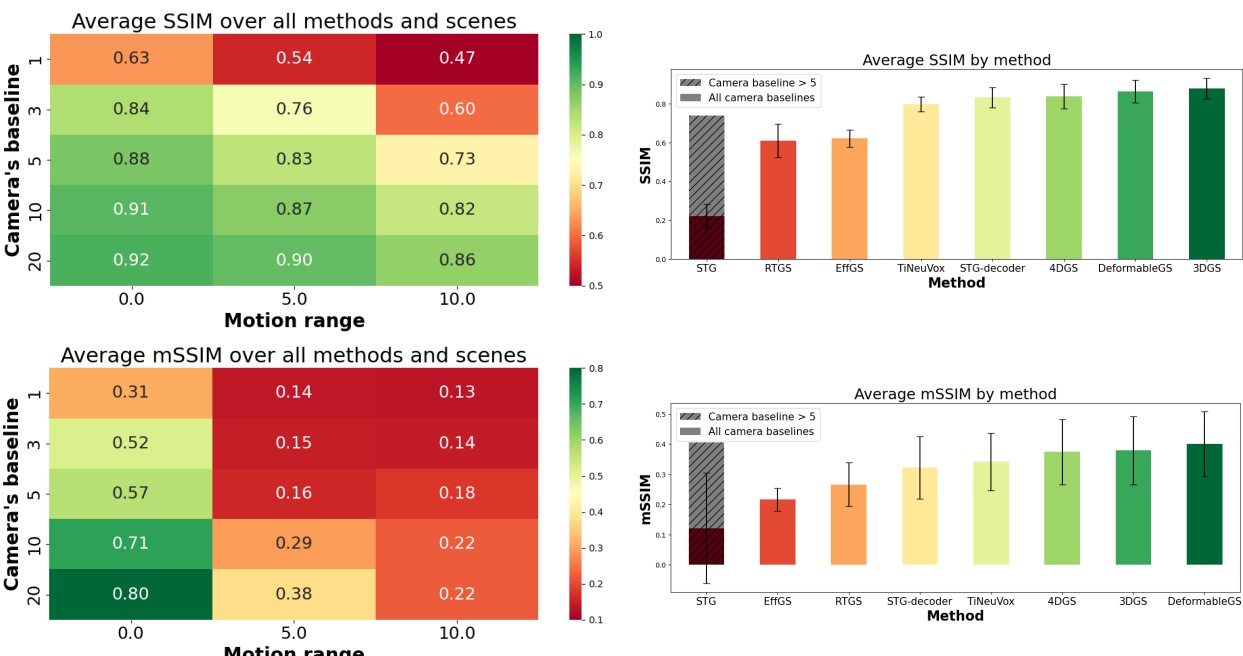

Figure 32: **Results on our instructive synthetic dataset.** *Top Left*: SSIM↑ heatmap shows how camera baseline and object motion range affect performance across all methods on average. Increasing object motion range (left to right) affects reconstruction performance negatively; decreasing camera baseline (bottom to top) has the same effect. *Top Right*: Ranking of methods by average SSIM over all scenes. Solid bars represent score computed over all scenes. Textured bar represents metrics computed on wide camera baselines. *Bottom Left*: Masked SSIM↑ heatmap: with the focus on dynamic object reconstruction, the metrics become worse on average. *Bottom Right*: Ranking for masked SSIM.

### A.3.4 MS-SSIM

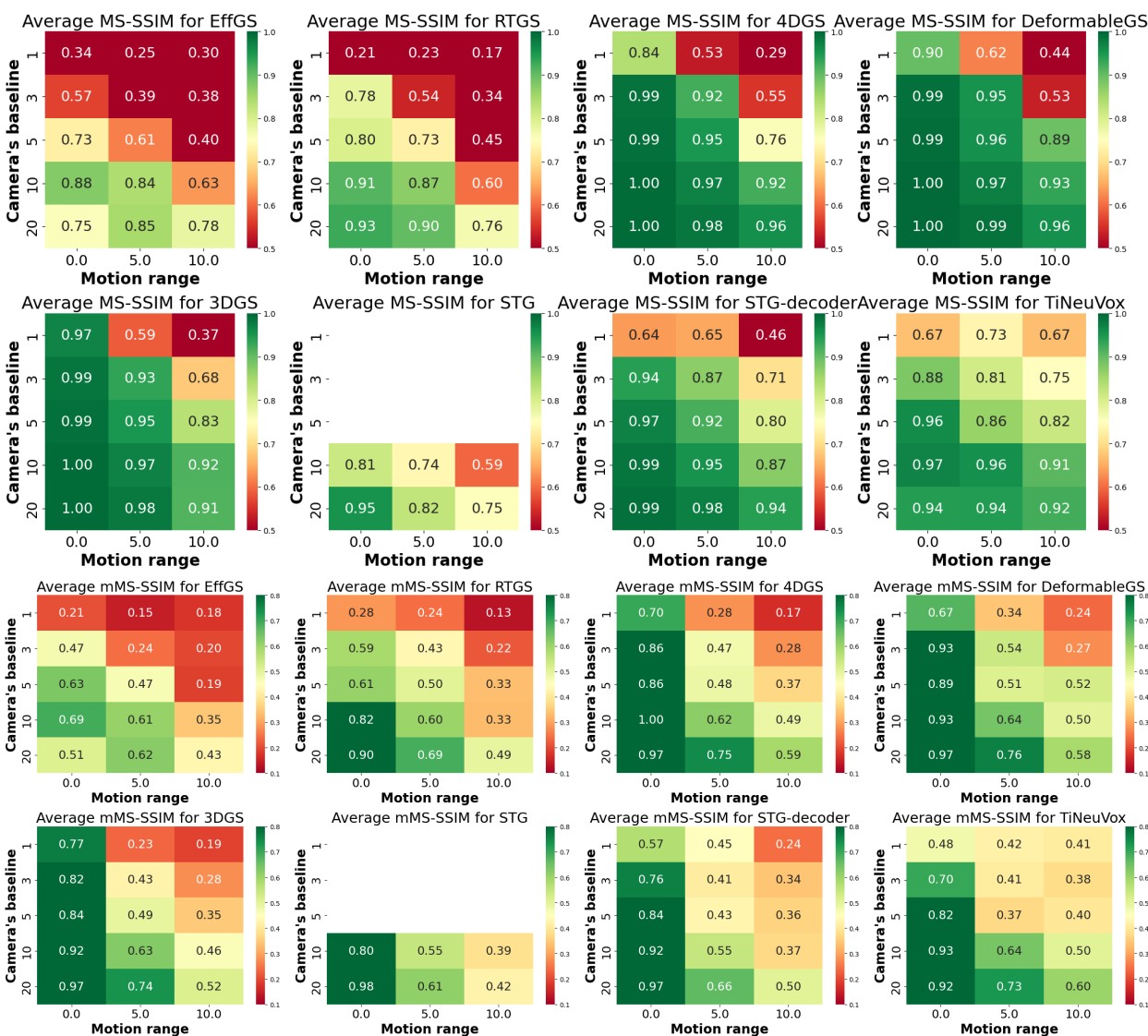

Figure 33: **Individual ablations for the instructive dataset, measured by MS-SSIM.** The figure shows camera baseline and motion range ablations for each method separately. *Top Two Rows:* MS-SSIM↑ heatmaps. *Bottom Two Rows:* mMS-SSIM↑ heatmaps. On average the reconstruction becomes harder with increase of the motion range (left to right) and with decrease of the camera baseline (bottom to top). Missing values indicate a non-converging model.

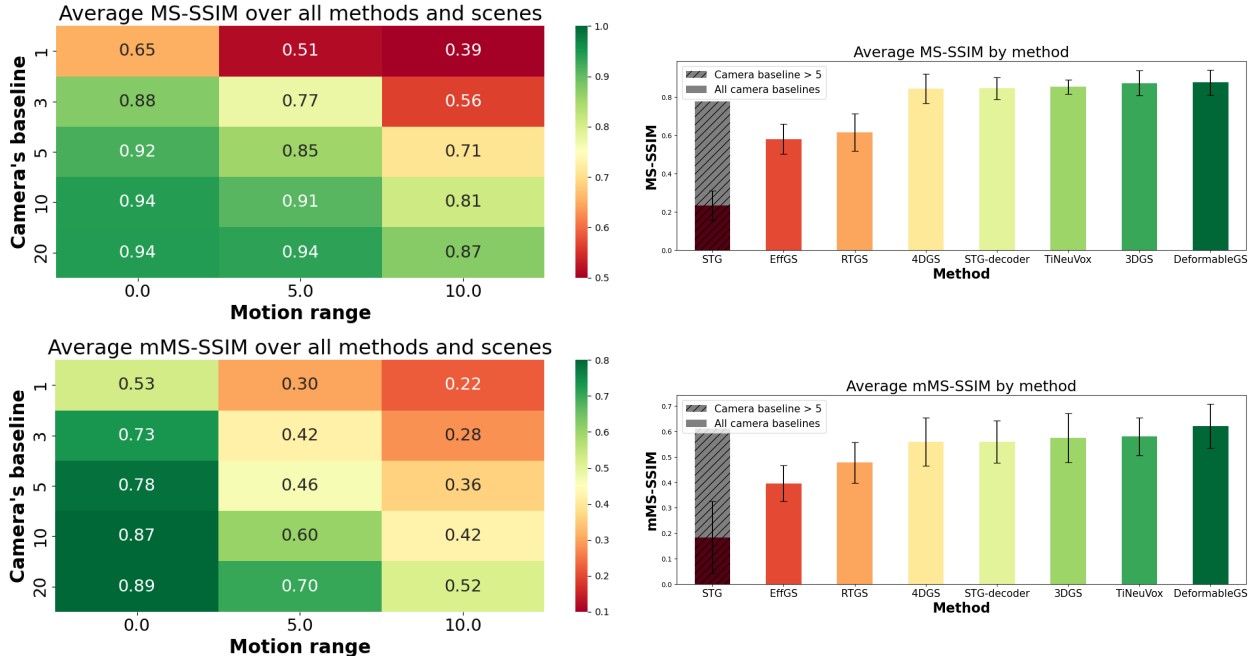

Figure 34: **Results on our instructive synthetic dataset.** *Left*: MS-SSIM↑ heatmap shows how camera baseline and object motion range affect performance across all methods on average. Increasing object motion range (left to right) affects reconstruction performance negatively; decreasing camera baseline (bottom to top) has the same effect. *Right*: Ranking of methods by average MS-SSIM over all scenes. Solid bars represent score computed over all scenes. Textured bar represents metrics computed on wide camera baselines. *Bottom Left*: Masked MS-SSIM↑ heatmap: with the focus on dynamic object reconstruction, the metrics become worse on average. *Bottom Right*: Ranking for masked MS-SSIM.

# B Additional Data-related Details

Our work shows a greater set of comparisons than existing works as they typically only report on a subset of the data by selection (Table 8).

Table 8: Previous works typically only report on a subset of data.

|  | Method | D-NeRF | Nerfies | HyperNeRF | NeRF-DS | iPhone |
|---|---|---|---|---|---|---|
| RTGS | Yang et al. (2024) | ✓ | ✗ | ✗ | ✗ | ✗ |
|  | Ours | ✓ | ✓ | ✓ | ✓ | ✓ |
| DeformableGS | Yang et al. (2023) | ✓ | ✗ | ✗ | ✓ | ✗ |
|  | Ours | ✓ | ✓ | ✓ | ✓ | ✓ |
| 4DGS | Wu et al. (2023) | ✓ | ✗ | 4 of 13 | ✗ | ✗ |
|  | Ours | ✓ | ✓ | ✓ | ✓ | ✓ |
| EffGS | Katsumata et al. (2023) | ✓ | ✗ | 4 of 13 | ✗ | ✗ |
|  | Ours | ✓ | ✓ | ✓ | ✓ | ✓ |
| STG | Li et al. (2023a) | ✗ | ✗ | ✗ | ✗ | ✗ |
|  | Ours | ✓ | ✓ | ✓ | ✓ | ✓ |

### B.1 Dataset Descriptions

D-NeRF (Pumarola et al. (2020)) is a synthetic dataset containing eight single-object scenes captured by 360-orbit inward-facing cameras. Test views are rendered at unseen angles. The test set is misaligned in the *lego* sequence, and so we use *lego*'s validation set for testing instead following Yang et al. (2023).

Nerfies (Park et al. (2021a)) and HyperNeRF (Park et al. (2021b)) contain general real-world scenes of kitchen table top actions, human faces, and outdoor animals. Thus far, most methods (Wu et al., 2023; Fang et al., 2022) report results on four sequences from the 'vrig' split of HyperNeRF dataset that uses a second test camera rigidly-attached to the first, rather than the 'interp' split to render held-out frames from a single camera (an easier task). Some works report their own *sequence* splits from HyperNeRF (Kratimenos et al., 2023). We include all 17 HyperNeRF sequences and 4 Nerfies sequences unless otherwise noted.

NeRF-DS (Yan et al. (2023)) contains many reflective surfaces in motion, such as silver jugs or glazed ceramic plates held by human hands in indoor tabletop scenes. This contains seven sequences. Test sequences are again generated by a second rigidly-attached camera. Lastly, the iPhone (Gao et al. (2022)) dataset's 14 scenes include large dynamic objects often undergoing large motions, with relatively small camera motions. Test views are from two static witness cameras with a large baseline difference.

We do not include the NVIDIA Dynamic Scene dataset (Yoon et al., 2020) because they sub-sample fake monocular camera sequences from a 12-camera wide baseline setup. This creates an unrealistically large degree of scene motion between frames.

### B.2 Erroneous Camera Pose

The HyperNeRF data has known bad camera poses. To quantify the camera pose error's effect on reconstruction quality, we improve the camera poses in HyperNeRF by segmenting out moving objects and highly specular regions and rerunning COLMAP; we report these experiments result in Appendix B.2.

Camera pose is required as input for most dynamic Gaussians methods for monocular videos. However, it's difficult to measure precise poses with real-world dynamic videos, leading to pose errors. Erroneous camera poses can be detected by simply applying static Gaussian Splatting Kerbl et al. (2023) to dynamic frames. If we observe blurry static background renderings as output, then this indicates erroneous camera poses.

HyperNeRF (Park et al., 2021b) is a dataset that suffers badly from camera pose inaccuracy, as dynamic regions were not masked for COLMAP (Schönberger & Frahm, 2016) during their camera pose estimation. To address this problem, we use SAM-Track (Cheng et al., 2023) plus human labor to mask dynamic, reflective and textureless regions, and then rerun COLMAP with these masks and customized arguments for each scene. Figure 35 shows improved static Gaussian rendering in background areas. Out of HyperNeRF's 17 scenes, we improve the camera poses in 7 scenes and slightly improve them in 3 scenes. We evaluate the pose quality by reporting mPSNR of static 3DGS on static regions, using the inverse of our motion masks, as summarized in Table 9.

We use **No Diff/Worse** group's 7 sequences from fixed and corresponding 7 sequences from original HyperNeRF dataset to show the effect of camera pose inaccuracy on reconstruction (Figures 36, 37, 38, 39). Performance is improved after pose fixes across for **EffGS**, **STG**s and **RTGS**. But, even though static regions are now deblurred in the standard 3DGS rendering showing improved poses, surprisingly not all method improve: **TiNeuVox**, **DeformableGS**, and **4DGS** instead often become worse.

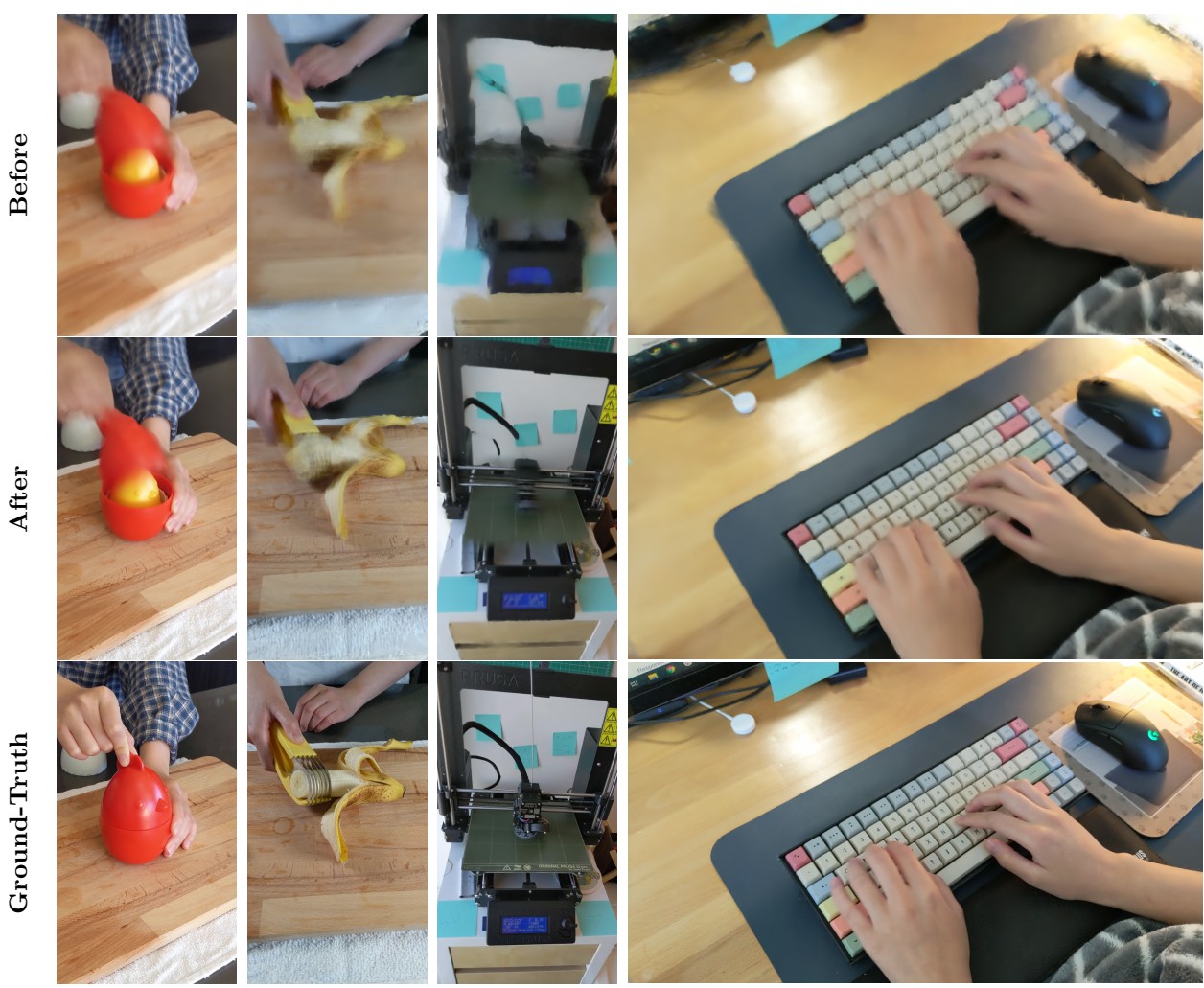

Figure 35: Comparison of ground-truth and static Gaussian Splatting (Kerbl et al., 2023) results on before and after correcting the camera poses for HyperNeRF (Park et al., 2021b) scenes (vrig-chicken, slice-banana, vrig-3dprinter, and keyboard).

| Result | Ratio Used | Scene | Original Poses mPSNR | Corrected Poses mPSNR |
|---|---|---|---|---|
| **Better** | 2x | chickchicken | 22.290 | 23.503 |
| | 2x | cut-lemon1 | 24.053 | 24.809 |
| | 1x | keyboard | 22.952 | 24.831 |
| | 2x | slice-banana | 22.794 | 24.690 |
| | 4x | tamping | 20.817 | 21.671 |
| | 1x | vrig-chicken | 23.334 | 26.075 |
| | 1x | vrig-3dprinter | 17.685 | 22.118 |
| **Slightly Better** | 2x | aleks-teapot | 22.343 | 22.543 |
| | 1x | broom2 | 21.987 | 22.810 |
| | 2x | cross-hands1 | 19.552 | 19.987 |
| **No Diff/Worse** | 1x | americano | 24.348 | 24.418 |
| | 1x | espresso | 22.707 | 22.291 |
| | 2x | hand1-dense-v2 | 25.547 | 25.214 |
| | 1x | oven-mitts | 25.565 | 19.891 |
| | 1x | split-cookie | 24.389 | 24.277 |
| | 2x | torchocolate | 23.345 | 18.356 |
| | 2x | vrig-peel-banana | 23.427 | 22.251 |

Table 9: Static region masked PSNR results for HyperNeRF scenes before and after correcting the camera poses.

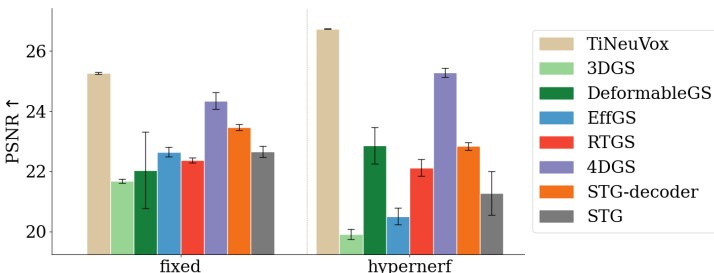

Figure 36: PSNR on the same set of HyperNeRF sequences to show Camera inaccuracy's effect on reconstruction. Higher is better.

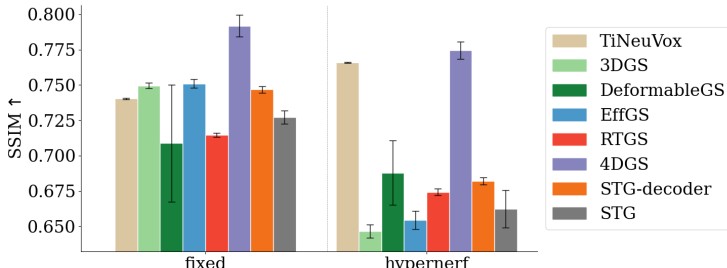

Figure 37: SSIM on the same set of HyperNeRF sequences to show Camera inaccuracy's effect on reconstruction. Higher is better.

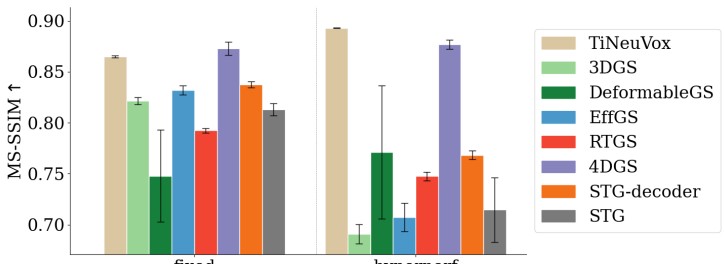

Figure 38: MS-SSIM on the same set of HyperNeRF sequences to show Camera inaccuracy's effect on reconstruction. Higher is better.

### B.3 Camera Pose Sensitivity Study on Instructive Dataset

We perform an additional study similar to Appendix B.2, where we introduce rotational noise into our instructive dataset's camera poses. Namely, we rotate the camera by $U[-\alpha, \alpha]$ where $\alpha \in \{1°, 5°\}$. We perform the study on non-rotating cube sequences.

Overall results of all methods in Table 10 indicates that there is a significant drop in quality ($>$3dB PSNR) between 0° noise and 1°. Further increasing the noise to 5° degrades most metrics as well, but not as dramatically. Qualitatively, we observe (see Figure 41) a jumping camera effect, that demonstrates the overfit happening due to the noise, and blurriness of the background.

Per-method analysis in Table 11 shows that although the noise largely does not affect the ranking of the methods by average performance (Figure 40), we observe a certain flattening of PSNR results, where the gap between best-performing methods like DeformableGS, and the worst ones tightens. At the same time, degradation for LPIPS and SSIM stays more uniform across the board.

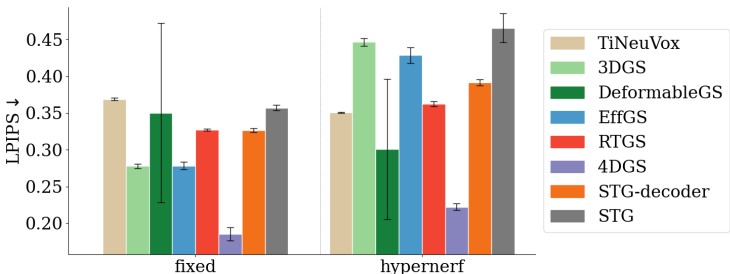

Figure 39: LPIPS on the same set of HyperNeRF sequences to show Camera inaccuracy's effect on reconstruction. Lower is better.

| Perturbation Angle | PSNR ↑ | Δ PSNR | SSIM ↑ | Δ SSIM | LPIPS ↓ | Δ LPIPS |
|---|---|---|---|---|---|---|
| 0° | 25.97 | 0.00 | 0.78 | 0.00 | 0.23 | 0.00 |
| 1° | 22.65 | -3.32 | 0.55 | -0.23 | 0.31 | +0.08 |
| 5° | 22.78 | -3.19 | 0.54 | -0.24 | 0.37 | +0.14 |

Table 10: Performance metrics at different perturbation angles with delta from baseline (0° noise).

| Method | Angle | PSNR ↑ | | SSIM ↑ | | LPIPS ↓ | |
|---|---|---|---|---|---|---|---|
| | | Value | Δ | Value | Δ | Value | Δ |
| DeformableGS | 0° | 29.41 | 0.00 | 0.87 | 0.00 | 0.12 | 0.00 |
| | 1° | 24.92 | -4.49 | 0.58 | -0.29 | 0.17 | +0.05 |
| | 5° | 24.66 | -4.75 | 0.57 | -0.30 | 0.21 | +0.09 |
| 3DGS | 0° | 27.71 | 0.00 | 0.87 | 0.00 | 0.16 | 0.00 |
| | 1° | 22.52 | -5.19 | 0.56 | -0.31 | 0.28 | +0.12 |
| | 5° | 22.61 | -5.10 | 0.53 | -0.34 | 0.38 | +0.22 |
| 4DGS | 0° | 27.26 | 0.00 | 0.84 | 0.00 | 0.16 | 0.00 |
| | 1° | 22.92 | -4.34 | 0.56 | -0.28 | 0.23 | +0.07 |
| | 5° | 23.98 | -3.28 | 0.56 | -0.28 | 0.24 | +0.08 |
| STG-decoder | 0° | 27.29 | 0.00 | 0.83 | 0.00 | 0.18 | 0.00 |
| | 1° | 24.24 | -3.05 | 0.58 | -0.25 | 0.27 | +0.09 |
| | 5° | 23.63 | -3.66 | 0.56 | -0.27 | 0.34 | +0.16 |
| EffGS | 0° | 22.55 | 0.00 | 0.63 | 0.00 | 0.36 | 0.00 |
| | 1° | 21.85 | -0.70 | 0.55 | -0.08 | 0.40 | +0.04 |
| | 5° | 22.42 | -0.13 | 0.55 | -0.08 | 0.45 | +0.09 |
| RTGS | 0° | 21.00 | 0.00 | 0.60 | 0.00 | 0.40 | 0.00 |
| | 1° | 19.22 | -1.78 | 0.47 | -0.13 | 0.49 | +0.09 |
| | 5° | 19.87 | -1.13 | 0.47 | -0.13 | 0.54 | +0.14 |
| STG | 0° | 7.51 | 0.00 | 0.21 | 0.00 | 0.80 | 0.00 |
| | 1° | 9.26 | +1.75 | 0.22 | +0.01 | 0.75 | -0.05 |
| | 5° | 5.55 | -1.96 | 0.14 | -0.07 | 0.87 | +0.07 |

Table 11: Performance metrics by method at different perturbation angles with delta from baseline (0° noise).

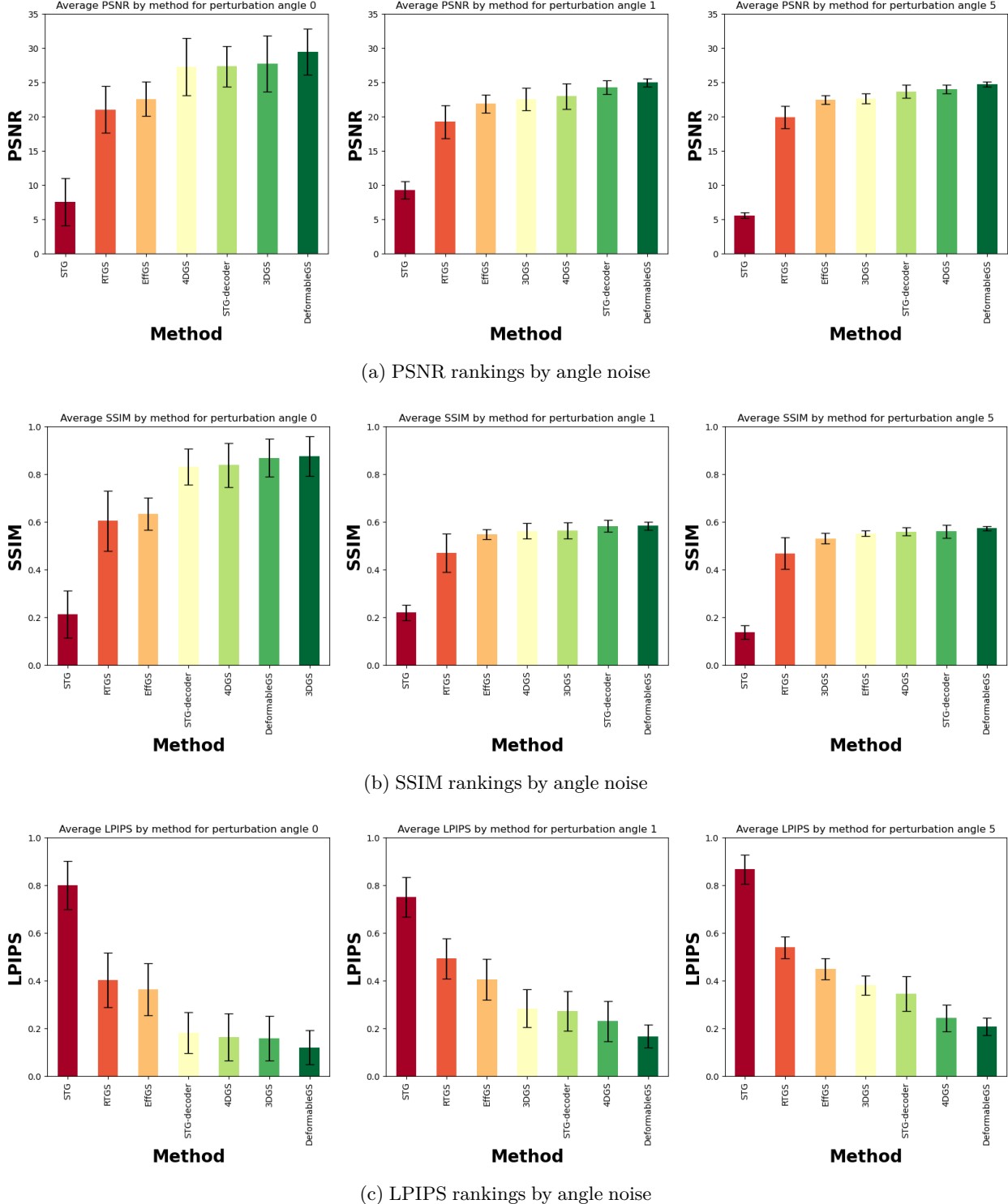

(a) PSNR rankings by angle noise

(b) SSIM rankings by angle noise

(c) LPIPS rankings by angle noise

Figure 40: **Rotational pose noise influence over methods ranking.** We observe that although the pose noise is influencing the absolute metrics, the rankings themselves are mostly unchanged.

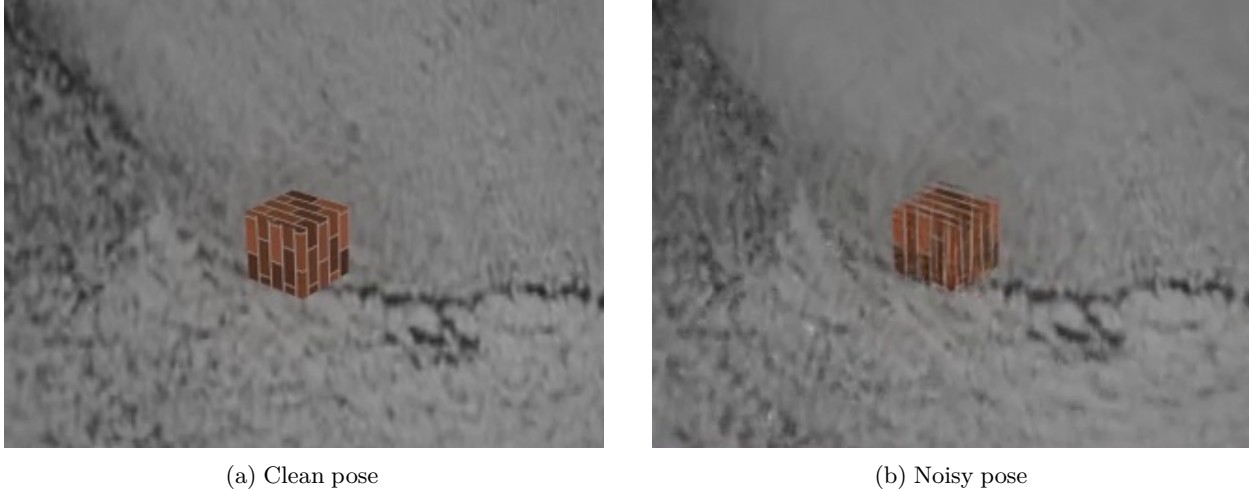

(a) Clean pose                                             (b) Noisy pose

Figure 41: **Qualitative effect of adding rotational noise to the pose.** Left: Reconstruction using clean pose. Right: Reconstruction using 5°-distorted poses. Both reconstructions are performed with DeformableGS.

## C    Experimental Implementation

### C.1    Shared Hyperparameters

Most of GS-related hyperparameters are inherited from 3DGS paper, and identical among algorithms with objective function (Equation (4)):

$$\mathcal{L} = \|I_{\text{pred}} - I_{\text{gt}}\| + \lambda_{\text{SSIM}} \cdot (1 - \text{SSIM}(I_{\text{pred}}, I_{\text{gt}})), \lambda_{\text{SSIM}} = 0.2 \tag{4}$$

For learning rates, people set spherical harmonics feature learning rate as 0.0025, opacity learning rate as 0.05, scaling learning rate as 0.005, rotation learning rate as 0.001, position learning rate as 0.00016 that exponentially decrease to 0.0000016 with learning rate delay 0.01, after 30000 iterations.

Gaussian densification starts from 500th training iteration, and ends until 15000th training iteration. Densification gradient threshold is set to 0.0002. The spherical harmonics (SH) degree is set to be 3 in 3DGS, percent dense is set to be 0.01.

To align with dynamic Gaussian works' implementation, a few hyperparameter might be different from 3DGS default setting:

Batch size 3DGS's default setting is to use batch size 1 during training. **RTGS** uses 4 for real-world scenes, **4DGS** uses 2 for real-world scenes, and **SpaceTimeGaussians** use 2 for all scenes.

Warm up stage **DeformableGS**, **4DGS** and **EffGS** first fix the motion component, and train the static part for the first 3000 steps.

Opacity reset During training, 3DGS set $G_i$'s opacity $\sigma_i$ to 0 periodically every 3000 iterations during densification phase. **4DGS** disable this opacity reset action when dealing with real-world sequences.

Initialization Original 3DGS suggests initialize $G_i$s with $100,000$ random points uniformly sampled in space for synthetic scenes, and with structure-from-motion (SfM) (Schönberger & Frahm, 2016) point cloud for real-world scenes. **RTGS** suggests appending extra uniformly-sampled random point cloud to SfM point cloud for a multi-view reconstruction dataset (Li et al., 2022). **EffGS** uses random point cloud in the place of SfM point cloud for the same dataset. We follow 3DGS's suggestion.

Apart from shared hyperparameters, each motion model also introduces extra hyperparameters.

### C.2    Extra Hyperparameters: Deformation

Deformation Learning Rate starts from 0.00016, and exponentially decays to 0.0000016 with delay multiplier 0.01 after certain steps. Deformation happens on position $x_i$, scale $s_i$ and rotation $q_i$, with $\sigma_i, c_i$ unchanged across time.

**DeformableGS** (Yang et al., 2023) Following original paper, we set network depth as 8; network width as 256; time embedding dim as 6 for synthetic, 10 for real-world; position embedding dim as 10; time is additionally processed with an embedding MLP with 3 layers, 256 width before feed into deformation network for synthetic scene; exponential decay max step as 40000.

**4DGS** (Wu et al., 2023) Following original paper, we set network depth as 0 for synthetic, 1 for real-world; width as 64 for synthetic, 128 for real-world; plane resolution as $[1, 2]$ for synthetic, $[1, 2, 4]$ for real-world; exponential decay max step as 30000; plane TV loss weight as 0.0001 for synthetic, 0.0002 for real-world; time smoothness loss weight as 0.01 for synthetic, 0.0001 for real-world; L1 time planes loss weight as 0.0001; Grid Learning Rate similarly exponentially decay like Deformation Learning Rate, except for going from 0.0016 to 0.000016; grid dimension as 2, input coordinate dimension as 4, output coordinate dimension as 16 for synthetic, 32 for real, grid resolution as $[64, 64, 64, 100]$ for synthetic, $[64, 64, 64, 150]$ for real.

To stabilize training of deformation networks, we perform gradient norm clipping by 5.

### C.3    Extra Hyperparameters: Curve

**EffGS** (Katsumata et al., 2023) uses Fourier Curve for $x_i$ motion, and Polynomial Curve for $q_i$ motion. Curve order is set to 2 for $x_i$, 1 for $q_i$. An optical-flow-based loss is additionally introduced to augment eq. (4) only for multi-view dataset, and here we do not include the flow loss as default.

**STG** (Li et al., 2023a) Original code is only supporting multi-view dataset as Gaussians are initialized by per-frame dense point cloud. To extend to monocular case, we copy point cloud for 10 times along time axis uniformly. During rendering, one option is to directly rasterize SH appearance as in 3DGS, but another

option is to rasterize feature instead, which would be later fed into a decoder network $D$ to generate color image. $D$'s trained alongside with learning rate 0.0001.

### C.4 Details of Dynamic 3D Gaussian Implementation

We update the codebase from Luiten et al. (2023) to support our evaluation across datasets. We use the HyperNeRF scenes with our improved cameras poses to minimize the reconstruction error of static areas.

# D Mathematical Motion Model Definitions

### D.1 Polynomial

Curve methods (Katsumata et al. (2023); Lin et al. (2023); Li et al. (2023a)) may use a polynomial basis of order $L$ to define a $f$ over time that determines Gaussian offsets, with coefficients $a_{i,j}$, and time offset $t_i$: Here, we denote $g_i$ of $G_i$ as the subset of parameters to change:

$$f(i,t) = \sum_{j=0}^{L} a_{i,j}(t - t_i)^j$$

$$G_{i,t} = \sum_{j=0}^{L} a_{i,j}(t - t_i)^j + G_i$$

$$(5)$$

### D.2 Fourier

$f$ could also use a Fourier basis with coefficients $a_{i,j}, b_{i,j} j = 0^L$, and time offset $t_i$:

$$G_{i,t} = \sum_{j=0}^{L} (a_{i,j} sin(j(t - t_i)) + b_{i,j} cos(j(t - t_i))) + G_i \tag{6}$$

### D.3 RBF

$f$ could be a Gaussian Radial Basis function with scaling factor $c_i$ and time offset $t_i$

$$G_{i,t} = G_i \exp(-c_i(t - t_i)^2) \tag{7}$$

### D.4 RTGS's 3D Rendering

RTGS (Yang et al., 2024) induces 3D world from 4D representation by obtaining the distribution of a 3D position conditioned on a specific time $t$:

$$x_{i|t} = x_i[:3] + \Sigma_i[:3,3]\Sigma_i[3,3]^{-1}(t - x_i[3]),$$

$$\Sigma_{i|t} = \Sigma_i[:3,:3] - \Sigma_i[:3,3]\Sigma_i[3,3]^{-1}\Sigma_i[3,:3]$$

$$(8)$$

# E    Related Work

Dynamic scene reconstruction has been a longstanding problem in computer vision (Newcombe et al., 2015; Slavcheva et al., 2017; Li et al., 2023b; Luiten et al., 2023). Earlier works utilize accurate depth cameras (Newcombe et al., 2015; Slavcheva et al., 2017) and reconstruct a dynamic scene using a monocular RGBD video. On the other hand, Joo et al. (2014) propose to use MAP visibility estimation with a large number of cameras and correspondences from optical flow to reconstruct a dynamic scene. In contrast, the focus of our study is on using a single-view RGB video to reconstruct dynamic scenes without taking depth or correspondences as input.

Recently, neural radiance fields (Mildenhall et al., 2020) (NeRFs) revolutionized 3D reconstruction allowing for the reconstruction of a static 3D scene only given 2D input images. This success led to various extensions onto the dynamic setting, i.e. the 4D domain (Ramasinghe et al., 2024; Wang et al., 2021; Song et al., 2023; Li et al., 2021; Gao et al., 2021; 2022; Cao & Johnson, 2023; Fridovich-Keil et al., 2023; Bui et al., 2023; Jang & Kim, 2022) by treating time as an additional dimension in the neural field. Other relevant approaches include combining a 3D NeRF representation with a learned time-dependent deformation field (Liu et al., 2023; Fang et al., 2022; Park et al., 2021a;b; Johnson et al., 2023; Kirschstein et al., 2023; Wang et al., 2023; Tretschk et al., 2020) and learning a set of motion basis (Li et al., 2023b) to optimize a dynamic scene.

The more recent explosion of works representing scenes with Gaussians (Kerbl et al., 2023; Keselman & Hebert, 2022; Chen & Wang, 2024) such as 3D Gaussian Splatting (3DGS) (Kerbl et al., 2023) has led to a number of promising extensions to its scene representation into the 4D domain which is the main focus of study in our work. As discussed in the earlier sections, one option to extend static 3DGS is to model motion as a 3D trajectory through time, learning motion by iteratively optimizing for per-Gaussian offsets into the next frame (Luiten et al., 2023; Sun et al., 2024; Duisterhof et al., 2023). Methods also represent these 3D trajectories differently such as using explicit motion basis (Das et al., 2023; Katsumata et al., 2023; Lin et al., 2023; Li et al., 2023a; Yu et al., 2023) or sparse control points (Huang et al., 2023). Another line of works (Wu et al., 2023; Yang et al., 2023; Liang et al., 2023; Guo et al., 2024; Lu et al., 2024; Liu et al., 2024; Diwen Wan, 2024) learn a 3D Gaussians that live in canonical space and optimize a time-conditioned deformation network to warp Gaussians from canonical space to each timeframe. Lastly, a few works (Duan et al., 2024; Yang et al., 2024) directly model Gaussians in 4D, i.e. extending across space and time.

# F    Discussion on Additional Regularization for Monocular Scenes

Reconstructing 3D scenes from only a monocular video often faces severe ambiguities, especially when camera motions are small or objects move rapidly. Although our analysis in earlier sections suggests that the chosen motion representation strongly influences stability, the intrinsic under-constrained nature of monocular data remains a fundamental limitation. As a result, several recent works have introduced additional regularization or supervision signals to guide dynamic Gaussian splatting methods toward more stable solutions.

## F.1    Smoothness and Rigid Priors

When motion is unknown or unsteady, enforcing physically plausible constraints can help reduce ambiguity. For instance, Luiten et al. (2023) use a suite of smoothness losses that encourage nearby Gaussian points (determined by $k$-nearest neighbors) to move in a more rigidly consistent manner, preserving relative distances over time. Subsequent works (Kratimenos et al., 2023; Lin et al., 2023; Bae et al., 2024; Stearns et al., 2024; Wang et al., 2024) adopt similar ideas to curb erratic Gaussian transformations. Meanwhile, Wu et al. (2023) employ a total-variation-style regularization on their discretized field, and Lin et al. (2023); Lu et al. (2024) both penalize changes in Gaussian parameters across short time windows to encourage piecewise stationarity. Even in purely 4D models (Duan et al., 2024), such smoothness priors can be extended to higher dimensions by promoting local rigidity and entropy-based consistency in opacity.

## F.2    Additional Supervision Signals

Instead of relying solely on RGB reprojection loss, one can incorporate complementary constraints like optical flow, depth, or pixel/particle tracks. Although these signals must themselves be estimated from data—potentially introducing noise—they can nonetheless reduce ambiguity when monocular cues alone are insufficient.

**Optical Flow.** Methods such as Katsumata et al. (2023) and Gao et al. (2024) compare a rendered flow map to optical flow predictions (e.g., from RAFT (Teed & Deng, 2020)), adding a loss term that ties the 2D motion field to the 3D Gaussian motion.

**Depth Supervision.** Depth can be rasterized alongside color by projecting Gaussian splats onto a depth buffer. Wang et al. (2024) enforce an $\ell_1$ penalty between rendered and ground-truth depths, while Liu et al. (2024) incorporate ordinal depth loss to handle inconsistencies in monocular depth estimates. Such approaches can anchor Gaussians in 3D space more reliably, especially for scenes with limited camera coverage.

**Pixel/Particle Tracks.** Another variant is to align Gaussian motion with pixel-level correspondences. For instance, Stearns et al. (2024) introduce a loss matching 2D pixel trajectories inferred by a separate tracking method. Similarly, Wang et al. (2024) impose consistency between splat trajectories and user-defined or automatically derived point tracks, including those along the $z$-axis.

### F.3 Scope and Outlook

Although these regularization or supervision mechanisms can significantly bolster stability, their integration typically requires additional computational overhead and careful tuning. Inferred optical flow or monocular depth may contain its own noise and domain biases. As a result, these strategies must be adopted judiciously, balancing the risk of error propagation against the potential benefit of stronger geometric constraints.

In summary, while our work does not focus on adding external priors or supervision, these methods highlight promising avenues for addressing the inherent ambiguities of monocular dynamic reconstruction. They can offer valuable tools for future research seeking to further improve the reliability and fidelity of dynamic Gaussian splatting pipelines in challenging, real-world environments.

