# OpenReview forum: "Monocular Dynamic Gaussian Splatting: Fast, Brittle, and Scene Complexity Rules"
_TMLR — Accepted by TMLR_

### Review · Reviewer_oxJq · 2025-02-01

**Summary Of Contributions:**

This paper provides a benchmark of Dynamic Gaussian Splatting methods for monocular view synthesis, unifying several motion models (e.g., polynomial, Fourier, deformation fields, 4D Gaussians) in one codebase and testing them on 50 real-world scenes and a new synthetic dataset. Key findings show that simpler and low-dimensional motion models are more robust, adaptive densification often leads to overfitting, and strictly monocular data remains challenging. A voxel-based baseline (TiNeuVox) outperforms many GS methods in quality but is slow. The paper releases open-source code, improved camera poses, and segmentation masks to enable fair comparisons and guide future research.

**Audience:**

Yes

**Claims And Evidence:**

Yes

**Requested Changes:**

- For each dynamic GS method, list the default values ​​, any visual-specific tweaks, and major hyperparameters (e.g., motion-model degree, deformation-field depth, regularization loss weight, and learning rate).
- Either (1) Add synthetic noise to (1) known ground-truth camera poses (in a scene where you have an accurate pose) or (2) choose some real-world sequences with different levels for these errors and determine flexibility. For example:
  - Introduce random disturbances to rotation/translation (like a small Gaussian offset) on different quantities (e.g., 0.5 °, 1 °, 3 °).
  - Compare the final rendering quality (PSNR, SSIM, LPIPS, etc.) before and after the pose disturbances.
- Discuss potential strategies to improve stability or address ambiguity in monocular dynamic scenes. Examples may include:
  - Optical flow: incorporating a penalty term that encourages stability consistency with 2D optical flow.
  - Smoothness / rigid prior: Additional constraints on Gaussian deformation fields so that local neighborhoods move consistently.

**Strengths And Weaknesses:**

Strength
- The paper introduces a large-scale and integrated evaluation of several dynamic Gaussian Splating methods on diverse datasets, leading to fair and transparent cross-method comparisons.
- A new custom synthetic dataset separates major factors (camera baseline, object speed), publishing how each method reacts to these variables under controlled settings.
- The paper goes beyond the standard metrics to analyze the foreground performance, overfitting preferences, and the role of adaptive Gaussian density for insight into the stability and failure mode of each algorithm.
- Replicable results build upon their findings by releasing code, improved camera poses, and segmentation masks.
- The paper concludes that simple, low-dimensional motion models often perform more strongly and that purely monocular settings remain challenging.

Weaknesses
- The paper contributes to the existing dynamic Gaussian Splatting (approaches by integrating, evaluating, and comparing. However, it does not propose a new technique for dynamic splatting. This lack of a novel method can reduce the technical contribution of the paper, even if the benchmarking itself is valuable.
- Authors refine the camera pose for HyperNeRF scenes, which can blur the stable background, but they do not systematically measure how each dynamic GS method handles different degrees of pose error. An intensive study (e.g., adding synthetic disturbances to take advantage of scenes with known poses will clarify the strength to maintain uncertainties.
- Empirical findings of the paper show how different motion representations or adaptive mechanisms may fail. However, this does not present a theoretical examination of why some configurations become unstable in monocular settings, nor does it propose a new stabilization strategy (e.g., additional priors or losses).

---

> ### Author Response · Authors · 2025-03-04
> **Response to Reviewer oxJq (1/2)**
>
> **Lack of a novel method**
>
> We appreciate your comment regarding the novelty of our work. While our paper does not propose a new dynamic splatting technique, our primary goal is to establish a robust, transparent benchmark for the rapidly expanding set of monocular dynamic Gaussian Splatting approaches. Multiple methods have emerged in quick succession, each claiming superiority based on inconsistent or incomplete evaluations. We see a clear, controlled, and fair comparison as crucial for advancing the field. By pinpointing where existing methods succeed or fail, we hope to guide future research toward stronger theoretical underpinnings and novel stabilization strategies.
>
> TMLR explicitly welcomes survey and benchmarking papers that identify open problems or highlight new connections (see [TMLR FAQ](https://jmlr.csail.mit.edu/tmlr/faq.html)). Our work fulfills this role by clarifying the current landscape, calling attention to instabilities in certain monocular settings, and suggesting avenues for improvement. Previous TMLR papers with a similar survey or benchmarking emphasis include [1,2]. We believe our contribution, though not a new algorithm, will spur further innovation in the dynamic Gaussian splatting community.
>
> [1] G. Ganev, M.S.M.S. Annamalai, E. De Cristofaro. *The Elusive Pursuit of Reproducing PATE-GAN: Benchmarking, Auditing, Debugging.* TMLR 2025
> [2] Yulei Qin, Yuncheng Yang, Pengcheng Guo, Gang Li, Hang Shao, Yuchen Shi, Zihan Xu, Yun Gu, Ke Li, and Xing Sun. *Unleashing the power of data tsunami: A comprehensive survey on data assessment and selection for instruction tuning of language models.* TMLR 2024
>
> ---
>
> **Survey on stability strategies**
> Great suggestion. We now mention the potential of regularization and other stabilization strategies in the conclusion section, and refer readers to the short survey in newly added section Appendix F.
>
> ---
>
> **More Implementation Details**
>
> Thank you for highlighting the importance of clearly enumerating the default values and major hyperparameters for each dynamic GS method. We fully agree that these details are crucial for reproducibility and clarity.
>
> We already have those details in Appendix C, but failed to explicitly note where these parameters can be found. We now include this sentence in Section 3.3:
> > “Implementation details (e.g., motion-model degree, deformation-field depth, regularization loss weight, and learning rate) can be found in Appendix C.”

---

> ### Author Response · Authors · 2025-03-04
> **Response to Reviewer oxJq (2/2)**
>
> **Pose error study**
>
> This is a great suggestion. We have conducted the requested experiments and included the results in Appendix Section B.3. Given the short timeframe and limited computational resources, we performed ablations with the camera rotated by 1° and 5° on the non-rotating subset of our Instructive Dataset. While this subset is limited, it provides a representative analysis of how pose distortions impact model performance.
>
> Due to OpenReview’s constraints, we provide the quantitative results both in the tables below and in the revised paper, and refer reviewers to the revised paper’s figures for qualitative comparisons.
>
> OpenReview-Table 1 & Paper-Table 10 indicates a significant drop in quality (>3dB PSNR) between 0° and 1° perturbation. Increasing the noise to 5° further degrades performance, though not as dramatically.
> Qualitatively (see Figure 41), we observe a jumping camera effect and a blurriness of the background most likely happening due to overfitting to the noisy poses
>
> A per-method analysis in OpenReview-Table 2 & Paper-Table 11 reveals that, while pose noise does not drastically alter method rankings (see Figure 40), it reduces the PSNR gap between top-performing methods (e.g., DeformableGS) and weaker ones. Meanwhile, LPIPS and SSIM degrade more uniformly, suggesting a consistent loss in perceptual quality across all methods.
>
>
> Table 1. Performance metrics at different perturbation angles with delta from baseline (0°).
> | Perturbation Angle | PSNR ↑  | Δ PSNR  | SSIM ↑  | Δ SSIM  | LPIPS ↓  | Δ LPIPS  |
> |--------------------|--------|--------|--------|--------|--------|--------|
> | 0°                | 25.97  | 0.00   | 0.78   | 0.00   | 0.23   | 0.00   |
> | 1°                | 22.65  | -3.32  | 0.55   | -0.23  | 0.31   | +0.08  |
> | 5°                | 22.78  | -3.19  | 0.54   | -0.24  | 0.37   | +0.14  |
>
>
> Table 2. Performance metrics by method at different perturbation angles with delta from baseline (0°).
> | Method        | Angle | PSNR ↑  | Δ PSNR  | SSIM ↑  | Δ SSIM  | LPIPS ↓  | Δ LPIPS  |
> |--------------|-------|--------|--------|--------|--------|--------|--------|
> | DeformableGS | 0°    | 29.41  | 0.00   | 0.87   | 0.00   | 0.12   | 0.00   |
> |              | 1°    | 24.92  | -4.49  | 0.58   | -0.29  | 0.17   | +0.05  |
> |              | 5°    | 24.66  | -4.75  | 0.57   | -0.30  | 0.21   | +0.09  |
> | 3DGS         | 0°    | 27.71  | 0.00   | 0.87   | 0.00   | 0.16   | 0.00   |
> |              | 1°    | 22.52  | -5.19  | 0.56   | -0.31  | 0.28   | +0.12  |
> |              | 5°    | 22.61  | -5.10  | 0.53   | -0.34  | 0.38   | +0.22  |
> | 4DGS         | 0°    | 27.26  | 0.00   | 0.84   | 0.00   | 0.16   | 0.00   |
> |              | 1°    | 22.92  | -4.34  | 0.56   | -0.28  | 0.23   | +0.07  |
> |              | 5°    | 23.98  | -3.28  | 0.56   | -0.28  | 0.24   | +0.08  |
> | STG-decoder  | 0°    | 27.29  | 0.00   | 0.83   | 0.00   | 0.18   | 0.00   |
> |              | 1°    | 24.24  | -3.05  | 0.58   | -0.25  | 0.27   | +0.09  |
> |              | 5°    | 23.63  | -3.66  | 0.56   | -0.27  | 0.34   | +0.16  |
> | EffGS        | 0°    | 22.55  | 0.00   | 0.63   | 0.00   | 0.36   | 0.00   |
> |              | 1°    | 21.85  | -0.70  | 0.55   | -0.08  | 0.40   | +0.04  |
> |              | 5°    | 22.42  | -0.13  | 0.55   | -0.08  | 0.45   | +0.09  |
> | RTGS         | 0°    | 21.00  | 0.00   | 0.60   | 0.00   | 0.40   | 0.00   |
> |              | 1°    | 19.22  | -1.78  | 0.47   | -0.13  | 0.49   | +0.09  |
> |              | 5°    | 19.87  | -1.13  | 0.47   | -0.13  | 0.54   | +0.14  |
> | STG          | 0°    | 7.51   | 0.00   | 0.21   | 0.00   | 0.80   | 0.00   |
> |              | 1°    | 9.26   | +1.75  | 0.22   | +0.01  | 0.75   | -0.05  |
> |              | 5°    | 5.55   | -1.96  | 0.14   | -0.07  | 0.87   | +0.07  |

---

### Review · Reviewer_mEie · 2025-02-04

**Summary Of Contributions:**

This paper introduces a comprehensive and systematic evaluation of recent dynamic Gaussian Splatting methods for the dynamic novel view synthesis task. The evaluation is conducted on 5 commonly-used datasets and 1 new synthetic dataset. The paper claims 8 findings through the systematic evaluation, ranging from the complexity of the motion model to the choice of Gaussian copy or splitting strategies, etc.

**Audience:**

Yes

**Claims And Evidence:**

Yes

**Requested Changes:**

I think the authors are encouraged to provide
1. A clear discussion about the telescoped monocular setting.
2. Conduct experiments on slightly more complex synthetic data.
3. Conduct experiments in a controlled manner and only change one setting (motion complex, motion field, specular objects, etc.) to demonstrate the claims.
4. Discuss more about the effectiveness of monocular depth estimation in DVS.

**Strengths And Weaknesses:**

Strengths:
1. This paper is well-written with a clear structure.
2. The paper studies an interesting and consolidated problem about the insufficient evaluation of dynamic Gaussian Splatting methods.
3. Some findings are inspiring, such as the problem caused by the adaptive density control

Weakness:
1. Some experimental settings are not so clear. The training sets of different datasets are not clear. As pointed out by DyCheck, most methods adopt the "telescoped" camera setting such that these methods use different cameras on different timesteps to create a pseudo monocular video. However, such a telescoped camera does not exist in the real world. Some discussions are missing here.
2. Synthetic data too simple. The proposed new synthetic dataset only involves a moving cube, which is too simplistic for real-world applications. It is not too difficult to involve some complex motions such as dancing avatars or using physics simulation.
3. Some claims are not well-supported with rigorous experiments. For example, "motion locality helps quality" is demonstrated by the comparison between "Deformable GS" and "EffGS". However, these two methods adopt different complexity in terms of motion function but not only in using per-gaussian representation or not. It's not so difficult to control all the variables here but only change one factor in experiments for demonstration.
4. Camera pose rectification is only conducted on HyperNeRF but not on the iPhone dataset. According to my experience, the iPhone dataset involves more severe camera pose errors, which may affect the accuracy of the conclusions.
5. The incorporation of monocular depth is a trending strategy but is not discussed. Since reconstructing 3D from a monocular video is too ill-posed with small camera motions, there is a trend to adopt recent accurate single-view depth estimation or video depth in the pipeline. It would beneficial to include more discussions.

---

> ### Author Response · Authors · 2025-03-04
> **Response to Reviewer mEie**
>
> **More experimental setting details**
>
>
> We appreciate the reviewer's comment on missing details in experimental settings. We think the reviewer means teleported, not telescoped, and after careful inspection, we believe this warrants additional clarification in Section 3.1. We added one sentence regarding dataset splits, and more regarding camera settings.
>
> ---
>
> **Synthetic data too simple**
>
> We appreciate the suggestion to incorporate more complex synthetic motions, such as dancing avatars or physics simulations. Our current choice of a single moving cube was deliberate: by focusing on a simple, rigid object and varying only camera baseline and motion range, we can isolate each factor’s impact on reconstruction. This simplicity is also beneficial because it makes it harder for inaccuracies to be masked—bad reconstruction is more obvious when the scene and motion are minimal.
>
> While we plan to explore more complex synthetic scenes in the future, our study already includes wide-ranging real-world datasets (e.g., NeRF-DS with specular objects, large-motion iPhone captures) that present diverse challenges in lighting, textures, and unstructured motion. Pairing a controlled synthetic scenario with these varied real-world scenes enables us to balance a clear factor-by-factor evaluation against the complexities encountered in practical applications.
>
> ---
>
> **“Motion locality helps quality” (Section 4.2)**
> Thank you for raising this concern. We would like to clarify that our conclusion about “motion locality” is not based solely on comparing DeformableGS with EffGS. In Section 4.2, we compare two “field-based” methods (DeformableGS and 4DGS) with three “per-Gaussian” methods (EffGS and the STG variants). While these methods do differ in motion-function complexity and hyperparameters, the same trend—improved quality through local, collective motion fields—emerges across all of them. We acknowledge that fully isolating every factor would require re-tuning an explosion of hyperparameters for each model variant, which is not trivial. Instead, we chose to evaluate each method “as is” using their given hyperparameters within a shared framework, effectively conducting a meta-review of their motion models. The consistent improvement across different motion representations reinforces our broader conclusion that incorporating local motion fields can enhance reconstruction quality.
>
> ---
>
> **Camera pose rectification**
>
> Thank you for raising this concern. We would like to clarify that the camera-pose rectification for HyperNeRF was used only in a supplementary experiment (Appendix B.2), where we re-estimated camera poses to explore how improved pose accuracy affects reconstruction quality. Crucially, this rectification does not apply to any of the main HyperNeRF experiments, which rely on the original (potentially noisy) poses. Likewise, we made no pose corrections for the iPhone dataset, since we believe that real-world pose errors are part of the natural “in-the-wild” pipeline. Our primary goal is to evaluate each method under realistic conditions, where mild to moderate pose drift is typical. While inaccuracies in camera poses can indeed affect quantitative metrics, they also demonstrate how each method copes with everyday data-collection challenges. We agree that refining camera poses for the iPhone dataset might further illuminate certain behaviors and plan to explore such techniques in future work.
>
> The wording in Section B.1 may have accidentally implied a universal rectification, where it read:
> > “The HyperNeRF data has known bad camera poses. To quantify the camera pose error’s effect on reconstruction quality, we improve the camera poses in HyperNeRF by segmenting out moving objects and highly specular regions and rerunning COLMAP; see Appendix B.2.”
>
> We have moved this line to the start of Section B.2 and emphasized that these rectified poses are not used for the main paper results.
>
> Lastly, we recognize camera noise as an important topic in monocular dynamic reconstruction and added an additional section Appendix B.3 with controlled camera noise with our Instructive Dataset to analyze such noise's influence on reconstruction.
>
> ---
>
> **Monocular depth strategy survey?**
> Great suggestion. We now mention the potential of regularization in the conclusion section, and refer readers to the short survey in newly added section Appendix F.

---

### Review · Reviewer_uwmz · 2025-02-19

**Summary Of Contributions:**

The paper presents a thorough investigation into dynamic gaussian splatting methods from monocular video. By selecting representative methods with various motion models, the study provides comprehensive comparisons across multiple datasets, including a newly introduced synthetic dataset. The work also proposes several key findings that might be helpful for further research.

The key contributions are:
1. Comprehensive Benchmarking and Analysis. The paper sets up a new codebase for fair comparison of different methods across a range of datasets.
2. Valuable Insights from Benchmarking. For example, the study highlights that the adaptive density control can lead to increased overfitting and occasional optimization failures.
3. Systematic Categorization of Motion Representations. The paper gives a clear categorization of Gaussian splatting methods based on their motion representations, ranging from per-Gaussian motion models to field-based approaches and even 4D Gaussian representations.
4. Introduction of a Novel Synthetic Dataset. A new instructive synthetic dataset is introduced, allowing for fine-grained control in camera and object motion.

**Audience:**

Yes

**Claims And Evidence:**

Yes

**Requested Changes:**

1. **Equation (1) Correction**
   Update Equation (1) to include the missing index k. Maybe it reads:
   $$\prod_{k=1}^{j-1}(1-\alpha_k p_k(u,v))$$
   *Impact:* Would strengthen the work.
2. **Clarification of Motion Model Categorization (Section 3.3)**
   Could you please explain why EffGS is categorized as using a local low-order polynomial basis while STG, which also employs a 1st-order polynomial basis, is not considered in the same category.
   *Impact:* Would strengthen the work.
3. **Details on Implementation of the Single Codebase (Section 3.3)**
   Section 3.3 states, "To minimize differences in implementation except for the motion model, we integrate these algorithms into a single codebase." Does this mean that all other implementation differences have been standardized or minimized? If so, could you provide more details on how this was achieved—for example, which aspects of the original implementations were unified and how any remaining differences were handled?
   *Impact:* **Critical** to securing the recommendation.
4. **Clarification on Dataset Averaging (Section 4.1)**
   It would be better to change “a summary of the findings, averaged across 5 datasets” to explicitly list which datasets are included—especially noting whether the Instructive synthetic dataset is part of the evaluation.
   *Impact:* Would strengthen the work.
5. **Explanation of Discrepancy in Table 3 for RTGS Performance**
   Table 3: In the original RTGS paper, the authors compare RTGS with TiNeuVox on the D-NeRF dataset and report that RTGS significantly outperforms TiNeuVox. However, your results in Table 2 do not reflect such a pronounced advantage. Could you clarify the reasons behind this discrepancy? This explanation is critical, as it directly impacts the reliability of your Table 2 results and the related findings.
   *Impact:* **Critical** to securing the recommendation.
6. **Error Bar Details in Figure 5**
   Could you provide more information on how error bars were calculated, like the number of runs.
   *Impact:* Would strengthen the work.
7. **Reformatting Section 4.4 Paragraphs**
   It seems that the first and second paragraphs in Section 4.4 should be combined.
   *Impact:* Would strengthen the work.
8. **Further Explanation of Efficiency and Overfitting Claims (Section 4.4)**
   Section 4.4 states, "(1) the efficiency (in both optimization and rendering) can shift undesirably across scenes, (2) the risk of overfitting increases and (3) optimization occasionally fails completely." Could you provide a more detailed explanation of how Figure 6 supports these claims?
   *Impact:* **Critical** to securing the recommendation.
9. **Clarification on NeRF-DS Dataset Results (Section 4.4)**
    Section 4.4 states, "The only exception is in the NeRF-DS dataset, where we hypothesize that the reflective objects in that data cause issues for all methods, flattening the performance disparities between them." However, Appendix A indicates that the methods do not perform significantly worse on NeRF-DS compared to other datasets. Could you explain this discrepancy?
    *Impact:* **Critical** to securing the recommendation.
10. **Discussion on Motion Representation Effectiveness**
    Overall, it is challenging to identify which motion representation is clearly superior. While Section 4.6 provides a clear ranking of the different methods, it does not sufficiently explain why one representation outperforms another. A more detailed discussion of the underlying factors would be highly valuable.
    *Impact:* Would strengthen the work.
11. **Title Justification for "Smooth Motion Helps"**
    The title suggests that "Smooth Motion Helps," yet the paper does not clearly demonstrate or explain how smooth motion contributes to improved performance.
    *Impact:* Would strengthen the work.

**Strengths And Weaknesses:**

Strengths:
1. This work is valuable itself. It targets at the problem of lacking fair comparison for various homogenized methods and provides valuable insights through thorough benchmarking.
2. In general the paper is well-written and clearly clarifies most of its details and findings.


Weaknesses:
1. Some of the experiment settings need to be further clarifed and this may influence the plausibility of the findings. Please refer to the **Requested Changes**.
2. The findings of the work are extensive and impressive. While the work selects representative methods from different motion model groups, it does not conclusively determine which motion representation is most advantageous. I understand that a method may perform better on one dataset and worse on another, but I would appreciate a more detailed explanation of why certain methods excel. Specifically, is the performance difference directly related to the chosen motion representation? How does the motion representation influence the reconstruction quality? A deeper exploration of these questions would add significant value to the work.

---

> ### Author Response · Authors · 2025-03-04
> **Response to Reviewer uwmz (1/2)**
>
> **Discussion on Motion Representation Effectiveness**
>
>
> Thank you for your valuable feedback regarding the need for a deeper exploration of why certain motion representations outperform others and how these representations influence reconstruction quality. In the revised manuscript, we have added a dedicated Section 5 that directly addresses these points.
>
> - *Why certain methods excel*: We systematically show that scene factors—like camera baselines, fast-moving objects, and specularities—can overshadow the choice of motion model. Section 5 clarifies how each parameterization may yield an advantage in specific scenarios.
>
> - *Influence of motion representation on reconstruction*: We present a consolidated discussion of how representational complexity (e.g., polynomial vs. MLP-based deformation fields, or 3D vs. 4D splatting) affects both scene fidelity and training reliability. We also highlight interaction effects with adaptive density control, which can exacerbate overfitting or lead to catastrophic optimization failures.
>
> ---
>
> **Clarification of Motion Model Categorization (Section 3.3)**
>
> Thank you for this insightful question. While both EffGS and STG do use polynomial bases, their complete motion models differ:
>
> 1. EffGS uses a combination of a Fourier basis and a polynomial basis per Gaussian to model motion, with each basis directly operating on the motion parameters.
> 2. STG, on the other hand, uses a RBF as part of the motion model, alongside with polynomial basis.
>
> We updated Section 3.3 to make this distinction clearer, aligns better with Table 1, and to better explain our categorization criteria:
>
> - Low-Order Poly+Fourier
> - Low-Order Poly+RBF
>
> ---
> **Details on Implementation of the Single Codebase (Section 3.3)**
>
> Our goal is to minimize all implementation differences—beyond just the motion model—by consolidating the entire training and rendering pipeline into a single codebase. This ensures a fair comparison and simplifies maintenance. Below, we detail the unification process and how remaining model-specific differences are managed.
>
> - **Shared Training Pipeline**
>   - *Unified Trainer Class:* Rather than dividing data loading, optimization, adaptive density control, and rendering into separate modules (as in the original GS works), we built a single “trainer” class. This class orchestrates the steps for each training iteration, including
>     - Parameter Estimation: Estimating Gaussian parameters given the chosen motion model (and deformation model, if applicable).
>     - Rendering: Using the estimated parameters to render an image.
>     - Loss Computation: Computing loss against the ground-truth image (batch or single).
>     - Adaptive Density Control & Optimization: Potentially triggering adaptive density control, then calling an optimizer to update parameters.
>     - Learning Rate Scheduling: Updating the learning rate(s) as part of the optimizer step.
>
>
> - **Integrated Gaussian Parameter Management**
>   - *Central Parameter Storage:* Gaussian parameters are now member variables of the trainer class, rather than spread across different modules.
>   - *Support for Multiple Models:*
>     - When using 3D motion models (e.g., DeformableGS), a separate motion model is also stored as part of the trainer class. This keeps all parameters—both Gaussian and motion—under a single training loop.
>     - Each motion model can still have its own dedicated package for Gaussian rasterization and deformation, ensuring parity with the original code while benefiting from the shared pipeline.
>
>
> - **Consistent Optimization and Loss Computation**
>   - *Single or Dual Optimizers:* A single optimizer typically handles Gaussian parameter updates with its own learning rate schedule, plus a second optimizer if the motion model is separate.
>   - *Unified Loss Function:* We integrate all loss types into one function. Only the relevant losses are enabled based on arguments.
>
> - **Rendering and Additional Operations**
>   - *Shared Color Decoder (STG):* STG’s original color decoding logic is preserved, integrated at the end of rendering for those models needing it.
>   - *Gaussian Property Functions:* All property functions are methods of the trainer class, selectively invoked per model type.
>
> - **Maintaining Parity with Original Implementations**
>   - *Model-Specific Packages and Classes:* We retain certain packages and classes (e.g., diff-gaussian-rasterization) to ensure fidelity with the original codes.
>   - *Exclusive Parameters and Training Settings:* Each motion model still defines its unique set of parameters. The unified pipeline standardizes scheduling, loss computation, etc.
>
> ---

---

> ### Author Response · Authors · 2025-03-04
> **Response to Reviewer uwmz (2/2)**
>
> **Clarification on Dataset Averaging (Section 4.1)**
>
> This is a good suggestion! We have expanded the Section 4.1 sentence to:
> > “D-NeRF, Nerfies, HyperNeRF, NeRF-DS and iPhone Datasets”
>
> ---
>
> **Explanation of RTGS Results**
>
> RTGS uses scene-specific hyperparameters within its optimization that cause large changes in accuracy.
> To ensure fairness in our evaluation, we carefully verified RTGS’s reported performance using two approaches. First, we ran our code given RTGS’s D-NeRF-specific per-scene hyperparameters as referenced from their official implementation, and obtained results closely matching their reported metrics. Second, we ran their official code unmodified, using their provided evaluation scripts for PSNR, SSIM, and MS-SSIM, and confirmed that our reproduced numbers align well with theirs.
>
> | Method                                        |   PSNR     |  SSIM    | MS-SSIM |
> |--------------------------------------|------------|----------|-------------|
> | RTGS official code                       | 33.9443  | 0.9737  |   0.9846  |
> | Ours + per-scene hyperparams  | 33.0643  | 0.9713  |   0.9817  |
>
> The observed difference stems from our benchmarking protocol, which enforces a standardized evaluation across all methods by using general hyperparameter settings across scenes. Instead of tuning each method’s hyperparameters per scene, we follow the general hyperparameters provided in each paper. This ensures a fair comparison that highlights inherent representational strengths and weaknesses without excessive tuning. While RTGS, like other methods, benefits from per-scene hyperparameter optimization, extending this practice to all methods would lead to an explosive number of experiments, making large-scale comparisons impractical.
>
> We believe that a benchmark’s value lies in its ability to reveal differences in fundamental method design rather than reporting best-case performance for each scene. Our approach provides a transparent and unbiased comparison, preventing hyperparameter optimizations from overshadowing core algorithmic differences. While scene-specific tuning may yield higher numbers, it introduces confounding factors that obscure meaningful insights. By adopting a uniform evaluation framework, we enable researchers to assess the core capabilities of each method in a fair and reproducible manner.
>
>
>
> ---
>
> **Error Bar Details in Figure 5**
>
> That’s a good suggestion. We added a paragraph to Section 3.2 explaining:
>
> > “**Mean and Variance** For each experiment, we perform $3$ runs to capture performance variability. In our visualizations, the height of each bar represents the mean ($\mu$), while the error bars extend one standard deviation ($\sigma$) above and below the mean.”
>
> ---
>
> **Reformatting Section 4.4 Paragraphs & Efficiency/Overfitting Claims**
>
>
> Thank you for your suggestion on section 4.4. We restructured Section 4.4 to better connect the evidence with our claims about adaptive density control.
>
> We noticed that the original version used Figure 6 to support three key issues (efficiency variation, overfitting risk, and optimization failures) before making a convincing connection between number of Gaussians and efficiency/overfitting, which made it hard to follow.
>
> In the revised version specifically:
>
> 1. We first show how frequency content leads to varying numbers of Gaussians across scenes.
> 2. We then demonstrate how this variation impacts both training and rendering efficiency using Figures 7–9.
> 3. Next, we introduce Figure 6 to illustrate the extent of this variation.
> 4. Finally, we summarize the three main issues (efficiency variation, overfitting risk, optimization failures) and provide evidence for each.
>
> ---
>
> **Clarification on NeRF-DS Dataset Results (Section 4.4)**
>
> We apologize for the confusion. By “flattening,” we meant the *relative* gaps among methods shrink, not that absolute performance drops significantly. Reflective scenes challenge every method similarly, reducing the margin among them. We clarify this further in Section A.2.5.
>
> ---
>
> **Title Justification for “Smooth Motion Helps”**
>
> We sincerely thank the reviewer for this insightful observation. We have revised the title to:
> > “Monocular Dynamic Gaussian Splatting: Fast, Brittle and Scene Complexity Rules.”
>
> This more accurately captures our conclusions that these methods are fast but can be brittle under challenging conditions, and scene complexity ultimately dominates performance.

---

### Author Response · Authors · 2025-03-04
**General Response to All Reviewers**

We would like to express our sincere gratitude to all reviewers for the time and effort invested in evaluating our submission. We also thank you for recognizing key contributions and strengths of our work:

- **Comprehensive Benchmarking and Analysis**. Several reviewers noted our “thorough investigation into dynamic Gaussian splatting methods from monocular video” (Reviewer uwmz) and “comprehensive and systematic evaluation” (Reviewer mEie), unifying multiple motion models in one codebase and testing them across diverse datasets (Reviewer oxJq).

- **Valuable Insights** such as the drawbacks of adaptive density control (e.g., “increased overfitting and occasional optimization failures,” Reviewer uwmz).

- **Systematic Categorization of Motion**. Our categorization of methods—ranging from per-Gaussian motion models to field-based and 4D Gaussians—was acknowledged (Reviewer uwmz).

- **New Synthetic Dataset**. Reviewers appreciated our dataset that isolates camera baseline and object motion (Reviewer oxJq), enabling more fine-grained analysis (Reviewer uwmz).

- **Clarity and Structure**. The paper was deemed “well-written” (Reviewer uwmz) with a “clear structure” (Reviewer mEie), aiding understanding of our findings.

- **Resources for Replicability**. Our open-source code, improved camera poses, and segmentation masks (Reviewer oxJq) were commended for fostering fair comparisons and future research.

Your thorough reviews and thoughtful feedback have been invaluable in shaping the revisions to our paper. We have submitted a revised PDF and also individual responses, where we address your specific comments and suggestions in detail.

---

### Decision · Action_Editor_9qyK · 2025-05-06

**Recommendation:** Accept as is

**Comment:**

This paper presents a thorough comparative study and analysis of dynamic Gaussian splatting methods from monocular videos. The contribution is timely and provides valuable insights (e.g., the drawbacks of adaptive density control). The systematic categorization of motion and the new synthetic dataset offer fine-grained analysis that cannot be obtained by existing datasets.

The main weakness is that the paper tests individual algorithms as is, without factoring out individual design variables, potentially obscuring the analysis.

Overall, all three reviewers agree that benchmarking existing dynamic Gaussian splatting algorithms is a good contribution to the community. The authors have also addressed the reviewers' feedback sufficiently. The AE thus recommends to accept the paper.

**Audience:**

The TMLR's audience will be interested in learning the findings of this paper.

**Claims And Evidence:**

Yes, all the claims have been supported by comprehensive experiments.